# High-security learning-based optical encryption assisted by disordered metasurface

Zhipeng Yu[1,2,3,10], Huanhao Li [1,3,10], Wannian Zhao[2,10], Po-Sheng Huang[4], Yu-Tsung Lin[4], Jing Yao [1,3], Wenzhao Li[1,3], Qi Zhao [1,3], Pin Chieh Wu [4,5,6], Bo Li[2,7], Patrice Genevet [8] ✉, Qinghua Song [2,7] ✉ & Puxiang Lai [1,3,9] ✉

Artificial intelligence has gained significant attention for exploiting optical scattering for optical encryption. Conventional scattering media are inevitably influenced by instability or perturbations, and hence unsuitable for long-term scenarios. Additionally, the plaintext can be easily compromised due to the single channel within the medium and one-to-one mapping between input and output. To mitigate these issues, a stable spin-multiplexing disordered meta-surface (DM) with numerous polarized transmission channels serves as the scattering medium, and a double-secure procedure with superposition of plaintext and security key achieves two-to-one mapping between input and output. In attack analysis, when the ciphertext, security key, and incident polarization are all correct, the plaintext can be decrypted. This system demonstrates excellent decryption efficiency over extended periods in noisy environments. The DM, functioning as an ultra-stable and active speckle generator, coupled with the double-secure approach, creates a highly secure speckle-based cryptosystem with immense potentials for practical applications.

When light transmits through scattering media like ground glass, photons are multiply scattered due to the inherent inhomogeneous distribution of refractive index, and they travel randomly along different paths, forming speckles which are visually observed as alternately dark and bright grains[1]. While multiple scattering can scramble the input information, this seemingly randomness is actually governed by a deterministic transformation defined by the scattering medium that bridges the input information and the speckle pattern[2]. The use of speckles has garnered attention in the context of information encryption[3–5]. In the past decades, some strategies have been

developed to decrypt the information hidden in the speckles or manipulate the scrambled information, including wavefront shaping techniques[6–12] and speckle correlation based on the memory effect[13–15].

Recently, artificial intelligence, especially deep learning[5,16–22], has emerged as a powerful tool for decrypting the carried information (especially complicated information like human face) from speckles. Its database encounters few risks of leakage since it can be immediately deleted when the training is done, and the decryption process only needs the well-trained network but not the database. The database can therefore be well protected and unknown to users in

[1]Department of Biomedical Engineering, Hong Kong Polytechnic University, Hong Kong SAR, China. [2]Tsinghua Shenzhen International Graduate School, Tsinghua University, Shenzhen, Guangdong, China. [3]Hong Kong Polytechnic University Shenzhen Research Institute, Shenzhen, Guangdong, China. [4]Department of Photonics, National Cheng Kung University, Tainan, Taiwan. [5]Center for Quantum Frontiers of Research & Technology (QFort), National Cheng Kung University, Tainan, Taiwan. [6]Meta-nanoPhotonics Center, National Cheng Kung University, Tainan, Taiwan. [7]Suzhou Laboratory, Suzhou, China. [8]Physics Department, Colorado School of Mines, Golden, CO, USA. [9]Photonics Research Institute, Hong Kong Polytechnic University, Hong Kong SAR, China. [10]These authors contributed equally: Zhipeng Yu, Huanhao Li, Wannian Zhao. ✉e-mail: patrice.genevet@mines.edu; song.qinghua@sz.tsinghua.edu.cn; puxiang.lai@polyu.edu.hk

principle. The further extension of the application, however, is critically limited. To begin with, the use of conventional scattering media (CSM), such as ground glass diffusers, impose restrictions to practical applications due to inherent defects in existing studies. The instability of the optical properties[6,23] and limited memory-effect range of these CSM systems[24] could be critical drawbacks. In a conventional disordered medium, such as a ground glass diffuser, the thickness is much larger than the optical wavelength, leading to three-dimensional optical inhomogeneities. As a result, the accumulation of optical scattering becomes highly susceptible to environmental perturbations, and the sensitivity to these changes progressively intensifies with the thickness of the scattering medium[24]. When the optical scattering of the CSM or the system is altered due to the environmental perturbation, the initial status is hardly to be recovered and the data validity is corrupted. To make the system work again, one has to recollect the data to form a new database and re-train to update parameters of the neural network to make the network adapted to the new dataset, which is time consuming, laborious, and usually impractical for real-world applications. In addition, in nearly all reported works, one-to-one mapping between the input and output without any security key or any additional authorization will lead to easy interpretation of the encoded information for stealers during the prolonged operation.

To overcome the instability issue for speckle-based applications, a dielectric disordered metasurface (DM) that has a two-dimensional structure consisting of nano-pillars with a random phase profile has been proposed to replace the CSM[24]. Different from CSM, the DM can be rigorously treated as a single layer scattering medium with each titanium dioxide meta-pillar fixed on the fused silica substrate, and thus it is less sensitive to environmental perturbation[24]. Recently, DMs have been used in some research explorations including wavefront shaping to achieve large field of view and high-resolution imaging[24], complex optical-field computational imaging[25], optical encryption with meta-hologram[26], long-range ordered phase distribution[27], as well as learning-based multispectral image reconstruction[28]. In Ref. [24], the advantages of DMs in terms of stability and large memory effect were demonstrated through comparison with CSMs. Nevertheless, these features should be further explored in this work since the relationship among the plaintext, security key, and ciphertext is way more complicated in a totally different application scenario. In addition, other features of DMs, such as polarization multiplexing which provides more dimensional modulation of the light field to boost the security in the field of optical encryption, can be further utilized.

Therefore, in this study, the usage of a spin-multiplexing DM with random phase profiles is proposed together with the double-secure procedure. As an arbitrary polarization can be decomposed into two orthogonal polarization states (such as right- and left-handed circular polarization (RCP and LCP)) of different weights, output speckles from the spin-multiplexing DM vary with the polarization of the incident beam. In principle, the number of channels associated with the incident polarization is infinite by simply adjusting the polarization of the incident light. In other words, a single DM can serve as an actively tunable speckle generator in a passive manner, producing multiple polarization encryption channels with one medium. As a result, the information transmission channel and security are considerably improved with a spin-multiplexing DM. Furtherly, in the double-secure procedure proposed in this study, each phase profile of the input of the optical system is the superposition of a target phase pattern (plaintext) and QR phase code (security key), and thus plaintext is firstly encrypted by the security key in the input terminal. The input beam is then projected through the scattering medium, which is used for the secondary encryption of the plaintext, to form a speckle pattern (ciphertext). The speckle pattern is thus a synthesis of the plaintext and security key in each polarization encryption channel. During the decryption process, the ciphertext and the security key serve as the inputs of the neural network to decipher the plaintext, in which two-to-one mapping between the input and output of the optical system and the neural network are created to provide additional security to prevent unauthorized access to the encoded information.

In this work, we have employed multiple neural networks, denoted as $P(i)$-DMNet ($i = 1,2,3...$), with the same DM. Each network is individually trained using data collected with the corresponding polarization $P(i)$ of incident beams. The data independency for these polarizations has been confirmed through extensive data cross-validation. During our analysis of potential attacks, we observed that successful decryption of the plaintext can only occur when both the ciphertext and the security key (i.e., QR code) are correct and fed into the neural network with the matched polarization state. Any modifications or variations on the security key, such as alterations in speckles or QR code, can significantly diminish the decryption performance, proving eminent protection against hacking attempts. At last, the system is put in a noisy environment for a performance test over 135 h, during which the DM shows excellent large-noise tolerance with good decryption efficiency. Surprisingly, the DM exhibits supreme self-recovery ability to the initial state, which attributes to the large memory effect of the medium[24]. Such phenomenal performance avoids additional training for the network when new data is fetched or added. The proposed method provides high security for speckle-based optical encryption with robust performance, which opens new venues for many real-world applications[29].

## Results

### Working principle

The whole process can be divided into two stages: optical encryption and learning-based decryption, as shown in Fig. 1. In the optical encryption stage (Fig. 1a), the sender (Alice) projects a light beam of two different polarizations ($P(i)$ or $P(j)$) ($i \neq j$) onto a plaintext, which is firstly encrypted by a QR code phase pattern (security key) and then traveling through the DM as the secondary infilling of the plaintext, generating a speckle pattern (ciphertext). The DM scatters light differently with different input polarizations due to the spin-multiplexing random phase design. The relationship among the speckle, plaintext, security key, and DM can be expressed as:

$$U(x,y,z) = \iint U_P(x_0,y_0) U_S(x_0,y_0) U_{DM}(x_0,y_0) h(x-x_0, y-y_0, z)\, dx_0 dy_0,$$

$$(1)$$

where $U_P(x_0, y_0)$, $U_S(x_0, y_0)$, and $U_{DM}(x_0, y_0)$ correspond to the functions of the plaintext, security key, and DM, respectively, and $h(x, y, z)$ is an impulse response. From Eq. (1), it is very clear that the security key and the DM are applied encryption on the plaintext in sequence to achieve double-secure function. In addition, as $U_{DM}(x_0, y_0)$ varies with the change of incident beam polarization according to the design, multi-channel encryption can be implemented by changing the polarization of the incident beam.

In the learning-based decryption stage, several different deep neural networks (DNN) sharing the same structure, termed as $P(i)$-DMNet and $P(j)$-DMNet (Fig. 1b), are trained with data from incident beams of $P(i)$ and $P(j)$, in which ciphertext and the security key serve as the inputs to decode the plaintext. The receiver (Bob) needs authorization from Alice to acquire the security key and the polarization of the incident beam. Assuming that Bob can receive the ciphertext at the output terminal in real time by himself, he can directly get access to the plaintext by feeding the ciphertext and QR code into the polarization-matched network. For hackers who can even have access to the ciphertext, they cannot decrypt the plaintext without the authentication from Alice (i.e., lack of the security key and the polarization of the incident beam).

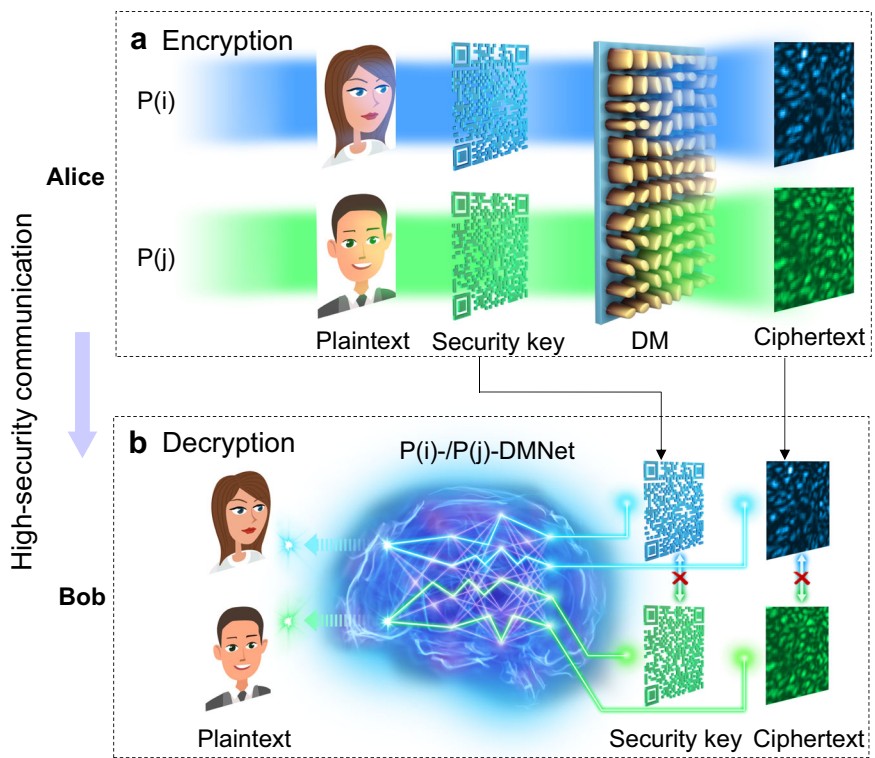

**Fig. 1 | Conceptional diagram of the proposed optical encryption system.**
**a** Optical encryption. The sender (Alice) illuminates light beams with two different polarizations of $P(i)$ and $P(j)$ onto the phase profiles of the superposition of plaintexts (human face images) and security keys (QR codes), which propagates through DM, generating ciphertexts (speckles). **b** Learning-based decryption. Two deep neural networks (DNN) of the same structure, e.g., $P(i)$-DMNet and $P(j)$-DMNet, are trained with data obtained with incident beams of $P(i)$ and $P(j)$, respectively. After recording the ciphertext and being authorized by Alice to acquire the security key and the polarization of the incident beam, the receiver (Bob) can feed the ciphertext and the security key into the corresponding neural network to decrypt the plaintext. The mark "×" above the straight line with arrows at both ends indicates that the information cannot be commutative. DM disordered metasurface.

## Design of the disordered metasurface

The DM consists of elliptical titanium dioxide (TiO$_2$) meta-pillars, as shown in Fig. 2a. The meta-pillars are 600 nm tall ($h$) and rest on a square lattice with a periodic constant ($P$) of 350 nm, and the design wavelength is 488 nm. The length of two axis ($u$ and $v$) of meta-pillars varies in the range of 70–320 nm, such that a controllable propagation phase $\phi_{\text{propagation}}$ is introduced for both LCP and RCP light beams. The simulated phase delays ($\varphi_{xx}$ and $\varphi_{yy}$) of the meta-pillar for two orthogonal linear polarizations ($x$ and $y$) versus lengths based on a commercial software Lumerical FDTD are shown in Fig. 2b. The propagation phase of the structure can be calculated from $\varphi_{xx}$ and $\varphi_{yy}$, i.e., $\phi_{\text{propagation}} = \arg\left((e^{1i*\varphi_{xx}} - e^{1i*\varphi_{yy}})/2\right)$ (more details are discussed in Supplementary Note 1). The birefringent meta-pillar is rotated with a rotation angle of $\delta$ that is able to perform circular polarization (CP) conversion $|L\rangle \rightarrow e^{i2\delta}|R\rangle$ and $|R\rangle \rightarrow e^{-i2\delta}|L\rangle$, i.e., the LCP and RCP beams are converted to the opposite spin with a geometric phase (or Pancharatnam–Berry (PB) phase) $\phi_{\text{geometric}}$ of $2\delta$ and $-2\delta$, respectively. The combination of the propagation phase and geometric phase enables the decoupling of RCP and LCP light at the designed wavelength for multiplexing wavefront modulation applications[30]. Given the desired phase of two orthogonal CP light $\phi_{\text{RCP}}$ and $\phi_{\text{LCP}}$, the required propagation phase and geometric phase at each meta-pillar can be calculated as[31]

$$\phi_{\text{propagation}} = \frac{(\phi_{\text{RCP}} + \phi_{\text{LCP}})}{2} \qquad (2)$$

$$\phi_{\text{geometric}} = \frac{(\phi_{\text{LCP}} - \phi_{\text{RCP}})}{4} \qquad (3)$$

Therefore, phase profiles of the DM for RCP and LCP incident beam are randomly distributed for the generation of speckle images.

Specific parameters of meta-pillar structures selected in the experiment can be found in Supplementary Note 2. As any polarization can be decomposed into two orthogonal polarization states (RCP and LCP in this study) with different weights[32], speckles generated from the DM vary with the polarization of the incident beam. A combination of a half-wave plate (HWP) and a quarter-wave plate (QWP) after the spatial light modulator (SLM) as shown in Fig. 3a is used to alter the polarization of the incident beam. Two specific orthogonal optical channels are defined by the two circular polarization states, i.e., $P(1)$: LCP and $P(7)$: RCP. In addition to these two orthogonal channels, 5 intermediate polarization channels, $P(2)$ to $P(6)$, located between $P(1)$ and $P(7)$, are created by rotating the QWP with an interval of 15°, as shown in the second row in Fig. 2c. Figures in the third row of Fig. 2c shows the recorded speckles corresponding to these 7 incident polarizations. Variation of Pearson correlation coefficient (PCC) of the speckles, taking the speckle of incident LCP as the reference, is illustrated in Fig. 2d. It can be seen that the speckle is highly sensitive to the rotation angle: the PCC gradually decreases from 1 to 0.08. Such a decrease of PCC can significantly impair the recovery efficiency of the input information. Meanwhile, it suggests the independence of each polarization state. It should be noted that only part of the diffused light field needs to be collected due to the complex mapping between the input and output light fields for information decryption[33], which further introduces benefits to the enhancement of the spatial security and the information capacity. Scanning electron microscope (SEM) images of the top and perspective views of the DM are shown in Fig. 2e (please refer "Methods" for more details).

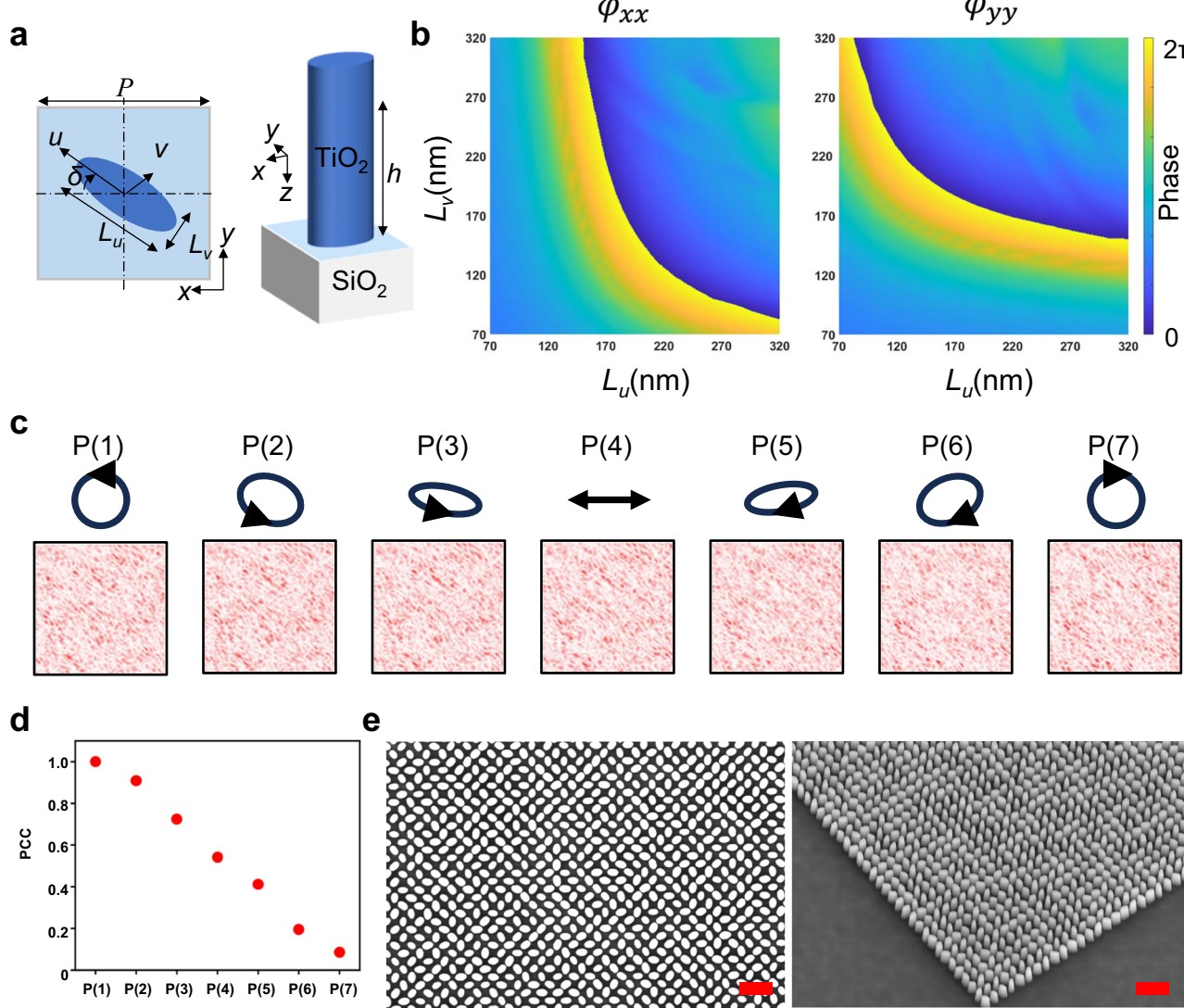

**Fig. 2 | DM structure and design. a** A TiO$_2$ unit meta-pillar of the DM with designed parameters is arranged in a square lattice on a fused silica substrate. **b** The simulated phase delays of the meta-pillar for two orthogonally linear polarizations (along $x$ and $y$ directions) versus lengths of the two axis of the DM. **c** Seven different polarization states between the LCP and RCP are defined by tuning the fast axis of QWP in the setup (Fig. 3a) and the recorded speckles corresponding to the 7 polarization states. **d** Speckle PCC versus polarization of incident beam, with the speckle associated with incident LCP as the reference. **e** Top (left) and perspective (right) views of SEM images of the fabricated DM. The scale bar in (**e**) is 1 mm. DM disordered metasurface, PCC Pearson's correlation coefficient, RCP right-handed circular polarization, LCP left-handed circular polarization.

## Decryption performance

The schematic diagram of the optical setup for data collection is illustrated in Fig. 3a. A collimated continuous-wave coherent laser beam with a wavelength of 488 nm (OBIS, Coherent, USA) is expanded to illuminate the aperture of a reflective SLM (HOLOEYE PLUTO VIS056, German), although a transmissive SLM for better visual observation is shown in Fig. 3a. Phase patterns are pre-loaded on the SLM to modulate the laser beam, which is polarized and tuned by a pair of a HWP and a QWP with controllable polarization state and then is slightly focused on the DM using a lens (L1) to generate optical speckles captured by a CMOS camera (FL3-U3-32S2M-CS, PointGrey, Canada). Another lens (L2) put in front of the camera is used to adjust the grain size of the recorded speckles. Sine the decryption is not a trivial inverse of the scattering process like other works[16,20,21] (more detailed discussion will be given in "Discussion"), a DNN named DMNet is specifically designed to match the physical process, with details provided in Supplementary Note 3.

When the training of DMNets in this experiment is done (more details can be found in Methods), the encryption process is ready. Notably, the DMNet trained and tested with the data generated via an RCP incident beam, i.e., $P(7)$ polarization in Fig. 2c, serves as the example in this part, i.e., the RCP-DMNet or $P(7)$-DMNet. As shown in Fig. 3, by feeding both the ciphertext (i.e., speckles in Fig. 3c) and the security key (i.e., the QR code in Fig. 3d) into the well-trained DMNet, decrypted images can be retrieved with high quality, as shown in Fig. 3e. Many fine features on the retrieved human faces can be identically mapped to the ground truth images (plaintext, Fig. 3b)[34]. Metrics for evaluation, as well, indicate excellent performance with averaged PCC = 0.941 and structural similarity index measure (SSIM) = 0.833. An example with PCC and SSIM as high as 0.97 and 0.93, respectively, as listed in the second column in Fig. 3. The network is therefore proved to accomplish accurate information reconstruction from the speckles. Nevertheless, such success depends on another two factors which strictly ensure the decryption: the second input (i.e., QR code used in this study) and the matched polarization between

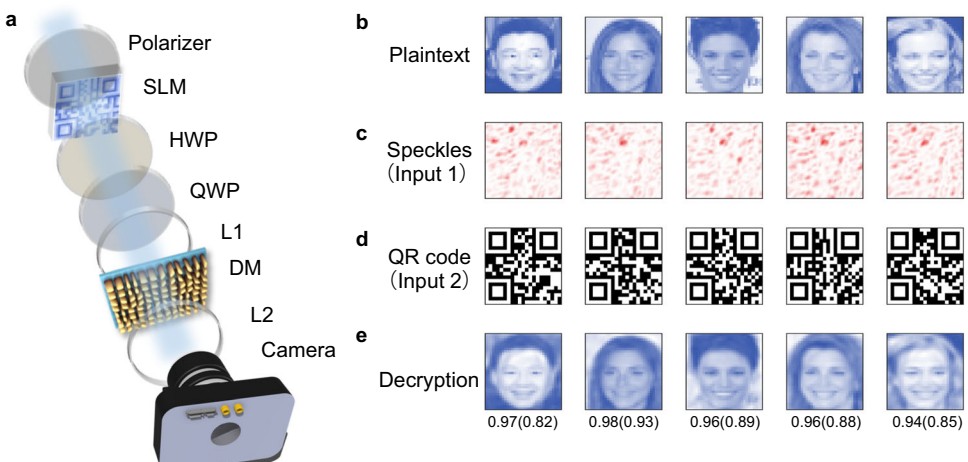

**Fig. 3 | Decryption performance based on DMNet. a** The schematic diagram of the optical setup. **b** Examples of plaintext for encryption. **c** The corresponding ciphertexts, i.e., the speckles. **d** Exampled QR codes. **e** The decrypted information by inputting (**c, d**) into the DMNet. The DMNet herein is trained by the RCP data. Inset numbers below each image in (**d**) are formatted as PCC(SSIM) between **b** the speckles and the network. ground truth and **e** the decrypted images. SLM spatial light modulator, DM disordered metasurface, HWP half-wave plate, L1, L2 lens, PCC Pearson's correlation coefficient, RCP right-handed circular polarization, QR quick response, QWP quarter-wave plate, HWP quarter-wave plate, SSIM structure similarity.

Other datasets such as fMNIST and Quickdraw (quantitative analysis of information complexity for different datasets can be referred to Supplementary Note 4) have also been tried, and the results can be referred to Supplementary Note 5.

## Attack analysis

**Security key protection.** As discussed in our previous work[21], speckle-based cryptosystem benefits from the complexity of the physical secret key demonstrating high-level security. Nevertheless, if the ciphertext (i.e., speckles) is accidentally obtained by the hackers, it is expected that the system still has the ability to protect itself. As designed in this study, additional authorized security key (i.e., the QR code) from the sender is needed for decryption at the receiver terminal. Several ciphertexts are generated when different QR codes (100 in this study) are paired up with each single plaintext. The performance of the decryption is therefore set to be sensitive to the change from the correct one in Input 2 in Fig. 3, given that the Input 1 or the ciphertext is correct. Likewise, RCP data serves as the example and five samples are randomly chosen for demonstrations, as shown in Fig. 4. As seen, if a uniform matrix is fed as Input 2 (Fig. 4aII), the DMNet merely outputs faces without recognizable features, whose PCC and SSIM (0.080 and 0.109, respectively) are both far below the performance with correct QR code (0.941 and 0.833, respectively; Fig. 4aI). Furtherly, excellent protection from the brutal attack for Input 2 is also achieved (Fig. 4aIII). By randomly generating one million binary-amplitude matrices to attack Input 2, the guessed plaintext is similar with that in Fig. 4aII. Notably, metrics to quantify the performance of brutal attack are not the average in Fig. 4bIII but the maximum, since the brutal attack succeed if one trial passes the guess regardless of its number of realizations. Nevertheless, the low PCC and SSIM (0.005 and 0.121, respectively) validate the safety of the designed network against the brutal attack for Input 2. Cases with mismatched pairs for the two inputs, for example, Input 1 is accurate but Input 2 is a "correct" QR code corresponding to another plaintext, can be found in Supplementary Note 6. The DMNet output (denoted as "Mismatched output") also fails to visualize the human faces but with similar patterns as shown in Rows II and III in Fig. 4a.

**Polarization channel protection.** In Fig. 2c, d, we have demonstrated the sensitivity of speckles to the incident polarization. Here, the data independency in these 7 polarization channels will be further verified.

Seven DMNets are individually trained using these seven polarized datasets, and each DMNet trained with $P(i)$ data is denoted as $P(i)$-DMNet ($i = 1,2,3,4,5,6,7$). With correct QR code (not shown in the Fig. 4c for simplicity), the plaintexts can only be correctly deciphered when the polarization state of the speckle matches that of the corresponding DMNet, as shown in the diagonal in Fig. 4c: $P(i)$ speckles are input into the $P(i)$-DMNet, resulting in decryption PCCs of -0.94. Once the polarization channels between the input data and network are mismatched, e.g., P(1)-speckles (LCP) input into P(7)-DMNet (RCP) or P(7)-speckles (RCP) input into P(1)-DMNet (LCP), the decrypted plaintext exhibits unrecognizable faces, with decryption PCCs of 0.0158 and 0.0268, respectively. In statistical analysis in Fig. 4d, it can be observed that the decryption PCCs for matched polarization states (-0.94 on the diagonal) are orders of magnitude higher than those with mismatched polarizations (<0.06 off the diagonal). That said, realizations for multi-channel decryption do not necessarily rely on the orthogonality of the polarization. The additional polarization states between the orthogonal ones can also support independence among the polarization channels. By jointly adjusting a half-wave plate and a quarter-wave plate, more polarization states can be created. In principle, arbitrary polarization state could be an encryption channel, with the polarization regulation as discussed in the "Working principle" section. Therefore, the feasibility of achieving multi-channel encryption, which requires independence of polarization channel and the realization of multi-polarization channels based on the DM, is assured.

## Stability test of the system

Stability of the decryption performance is critical in real applications but has seldom been discussed in earlier works due to the nature of CSM used in experiment. In this study, the system has been collecting data intermittently for 135 h (Periods 1–14 in Fig. 5a), whose status is characterized by the background PCC (blue dots). The background PCC is defined as the PCC between instant background speckle pattern and the initial one at Time = 0. All background speckle patterns are generated with the same uniform phase pattern displayed on the SLM as described in "Methods." Thereby, the initial status of the cryptosystem is defined in Period 1 in Fig. 5a, whose data is fed into RCP-DMNet for training with average decryption PCC (red bar) of around 0.94, as demonstrated in previous sessions. In other words, test data in the Periods 2–14 are new data for the network, which are collected under temporally varying medium status and have never been learned

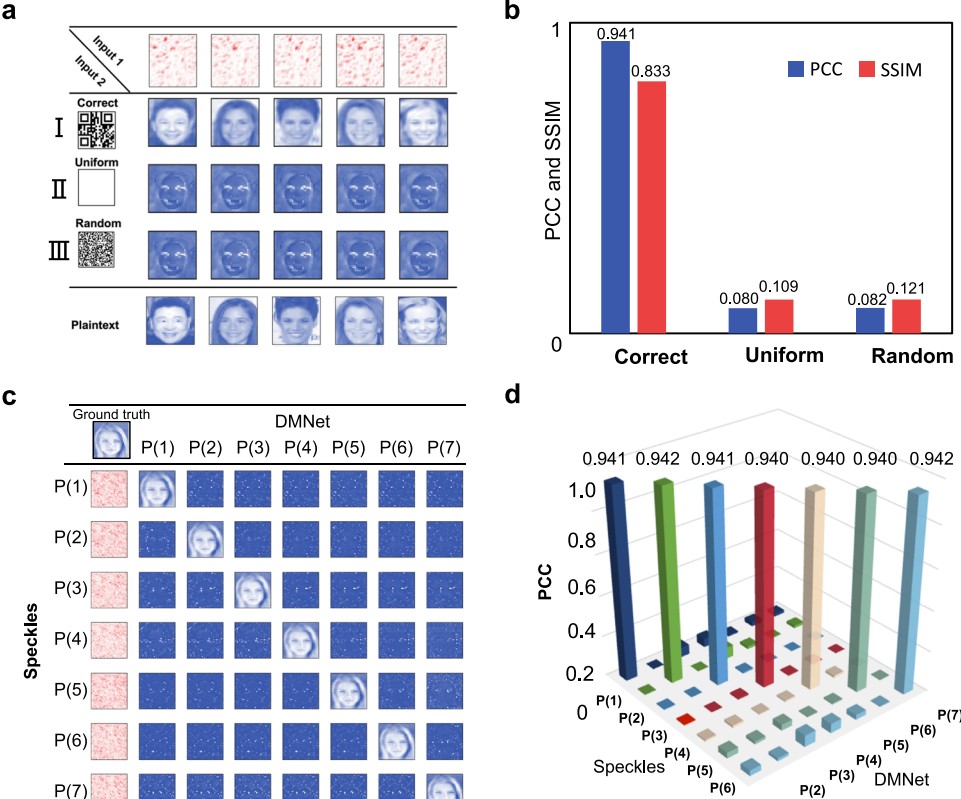

**Fig. 4 | Plaintext protection with double-secure scheme and multichannel encryption. a, b** Attack analysis regarding Input 2. Decryption with correct ciphertext (i.e., Input 1: speckles) by varying Input 2 with a correct QR code (Row I), a uniform pattern (Row II), and a random binary pattern (Row III) for **a** qualitative demonstration and **b** the statistics, quantifying the PCC and SSIM between the plaintext and decrypted images for Rows I–III. PCC Pearson's correlation coefficient, SSIM structure similarity index. The metrics for both "Correct" and "Uniform" are averaged over 2000 samples, and metrics for the 'Random' group is the average of 1,000,000 randomly generated binary-amplitude attacks. **c** Cross-validation for the decryption by inputting speckles with seven different polarization states (i.e., $P(i)$-speckles, $i$ = 1,2,3,4,5,6,7) into DMNet with seven different states (i.e., $P(i)$-DMNet, i = 1,2,3,4,5,6,7). (**d**) Averaged decryption PCC corresponding to the cross-validation arrangement in (**c**) and each is averaged over 2000 samples. QR quick response, PCC Pearson's correlation coefficient, SSIM structure similarity.

or probed by the network. Without additional training, decryption PCC in the following periods (Periods 2–14) changes accordingly with the background PCC, which is positively correlated. More importantly, the varying status can recover back to the initial status, e.g., Periods 2–6, Periods 7 to 8, and Periods 12–14, whose corresponding averaged decryption PCC recovers from 0.82 to 0.93, from 0.73 to 0.90, and from 0.68 to 0.90, respectively. The decrypted images can be seen in Fig. 5b. One should be noted that during such 135 h, the experiment is performed on the seventh floor and the environmental perturbations are general and diverse, including switching the laser/SLM/camera, other experiments on the same optical table, traffic around the building, large machine noise from adjacent machine room, etc. As seen, in our cryptosystem, the DM provides excellent stability against those everyday perturbations and the deviation from the initial status is reversible. Such a phenomenon can hardly be seen in CSM-based implementations (Ground glass diffuser, DG-10-220, Thorlabs) for such a long duration of time as shown in Fig. 5c: with everyday perturbations, the background PCC of the CSM-based system (with the same setup as the DM-based implementations) decreases obviously (down to around 0.2) without recovery back to the initial status. As seen in Fig. 5d, starting from period 2, the decryption performance also deteriorates over time. The fine facial features gradually erode, resulting in significant deviations from the ground truth images. This highlights an additional advantage of utilizing DM over CSM: for those media like ground glass diffusers, the deviation from the initial state is highly unpredictable and often irreversible. However, our proposed DM-based system exhibits reversibility (Fig. 5a). This remarkable

feature can be attributed to single-layered nature of the DM, which ensures a wider range of the memory effect[24]. This characteristic physically enables a more relaxed optical conjugation of the DM with the input wavefront compared to typical multi-layered diffusers. Therefore, our system can be practically recovered back to the initial status, as quantified by the background PCC of the recorded speckle (i.e., 0.98) when the perturbations become similar to those at initial status or when simply tuning the system is feasible. Furthermore, since no additional training for the network is needed over time, encrypting new plaintext with the proposed cryptosystem becomes practically feasible even though long period of time has elapsed since the network was trained.

## Discussion

In summary, we have proposed and demonstrated ultra-stable and high-security learning-based optical cryptosystem with a spin-multiplexing disordered metasurface. As speckles generated by the DM have polarization dependence on the incident beam, a single DM can support many polarization encryption channels. In the encryption process, each plaintext phase pattern is superimposed with a security key phase pattern serving as the input of the optical system to be transmitted through a DM to generate synthetic speckles in each polarization encryption channel. That is, the ciphertext (speckle pattern) contains two components originating from the phase object and the QR code, respectively. To decrypt and obtain the plaintext, the network needs to remove the contribution of the QR code from the speckle pattern, so that it can output the pure component of phase

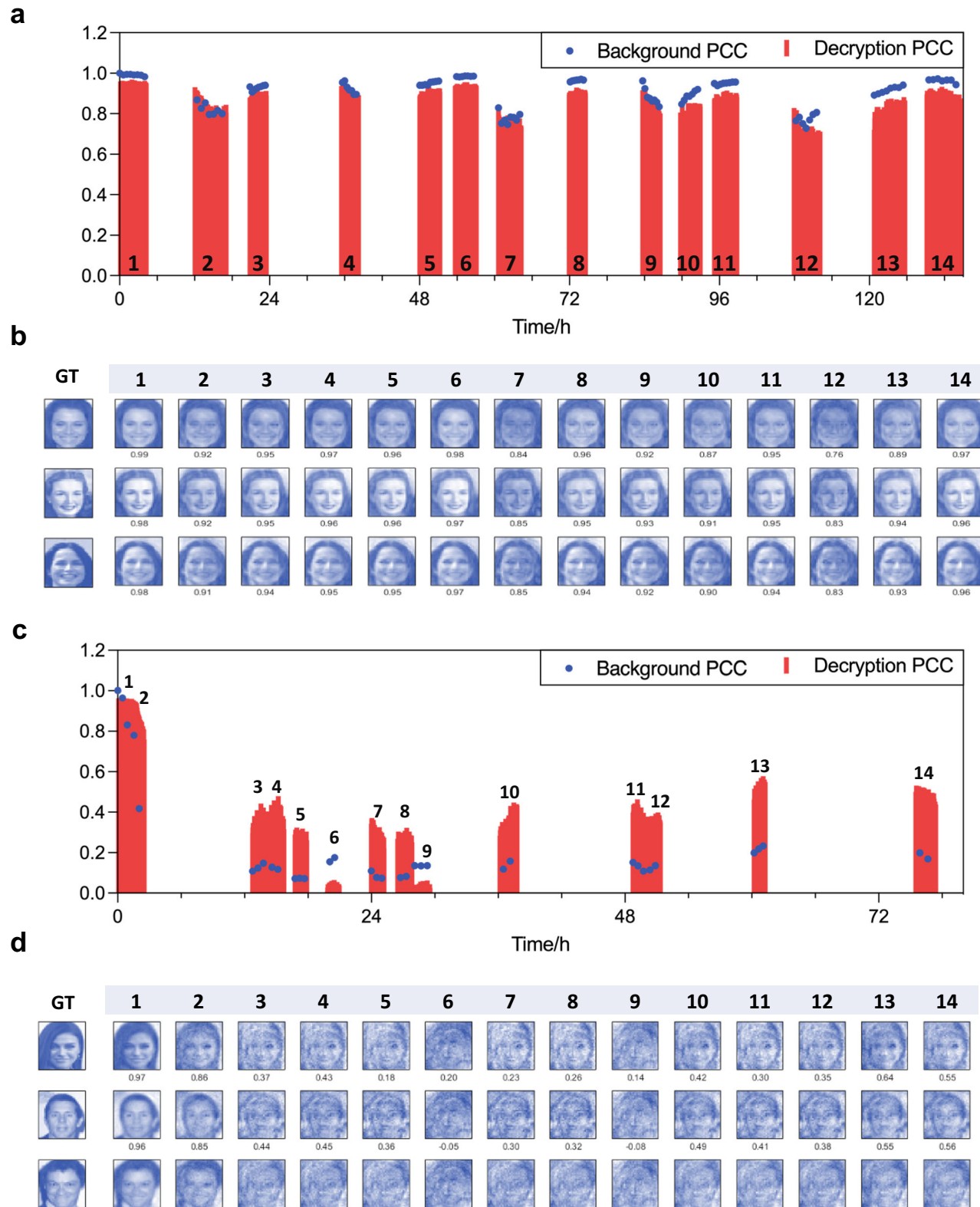

**Fig. 5 | Stability analysis for the decryption performance. a**, **b** Stability analysis for the DM-based decryption performance. **a** Background PCC (blue dots) and decryption PCC (red columns) based on the data collected in 14 periods. **b** Decryption performance for three representative examples with respect to the 14 periods in (**a**). Digits below each reconstructed images are the Decryption PCCs between the decrypted image and the ground truth image. **c**, **d** Stability analysis for the CSM-based decryption performance. **c**, **d** are the counterparts of (**a**, **b**), respectively, under the same experiment conditions with a ground glass to replace the DM as the scattering medium. GT ground truth, DM disordered metasurface, CSM conventional scattering medium, PCC Pearson's correlation coefficient.

object. The DMNet is, therefore, designed with two inputs (speckles and QR code) and one output (plaintext). By doing so, the decryption gains double-secure treatment with both speckles and QR code for correct decryption (Fig. 4a, b: mismatch of the QR code fails to decrypt the image). Such a scheme substantially differentiates the learning-based decryption process in our work from the peer works whose ciphertext (speckle pattern) contains only one component originating from the phase object, in which high-level security is hard to ensured[16,20,21]. In the decryption process, only when the speckle, the security key, and the polarization are all correct and matched, the plaintext can be deciphered correctly.

It also should be noted that in almost all of existing studies of optical encryption with metasurfaces, the information is encrypted into the metasurface structures and use them as physical carriers, which can be mainly divided into three categories: (1) utilizing single or multiple degrees of freedom of light such as wavelength[35,36], polarization[37,38], incident angle[39,40], nonlinear effect[41] and OAM[42]; (2) tunable modulation through external stimuli like chemical reactions[43], surrounding medium variation[44], mechanical actuation[45], electrical gating[46] and phase transition[47]; (3) other specific mechanisms such as computational ghost imaging with single-pixel imaging (SPI)[48], code division multiplexing[49] and secret sharing with cascaded metasurfaces[50]. The strategies in these works have limited the capacity and sharing of information as well as the compatibility with the mainstream digital information processing technologies. In our work, the information is encrypted onto output speckles instead of relying solely on metasurfaces themselves. Meanwhile, the DM can support multi-channel encryption, greatly enhanced information capacity and security, as well as improved compatibility.

The proposed system can also be equipped with some other encryption methods for further applications. The amount of QR codes in experiment is 100, and they can be numbered from "000" to "099" according to the carried information. A series of four-digit verification code is created by Alice (the sender) and the most significant digit of each verification code represents polarization of incident beam (taking two values: "0" represents RCP and "1" represents LCP) and remaining digits of each verification code represent the corresponding QR code. For example, if Bob receives a verification code "1088," he should use the QR code representing the serial number of "088" and the received LCP speckle as the input to be fed into the LCP-DMNet to decode the plaintext. Dynamic verification codes can also be created to further improve the security by changing the QR code regularly. Note that only one security key is present in this work, but, in theory, multi security keys can be added. For example, the plaintext can be divided into several parts and each part can be allocated with a security key and a specific polarization to further enhance the security.

In addition, the utilization of propagation phase inherently renders meta-pillars sensitive to the wavelength. In order to generate wavelength-sensitive speckles, meta-pillars with diverse correspondence between the propagation phase and wavelength are screened out[35]. Additionally, the sensitivity can be further enhanced by considering meta-pillars with increased height. Comprehensive simulation results regarding meta-pillars of different heights can be found in Supplementary Note 8.

Another important benefit of the DM is its capability to facilitate the mass production of custom-designed DMs with almost identical optical properties, including polarization sensitivity and wavelength sensitivity. Avoiding the required case-by-case medium characterization could be an essential asset in applying scattering-medium-based techniques. Moreover, the concept is applicable over a wide range of the electromagnetic spectrum from ultraviolet to terahertz light with a proper choice of low-loss materials for the meta-atoms[51–53]. Last, the employment of DM enables ultra stability and hence only one-time training for the network is needed even for new data acquired later, which is a breakthrough from previous speckle-based learning works.

With all these features discussed above, it is rational to expect that the versatile metasurface diffuser platform can be exploited for wide applications within and beyond speckle-based information encryption.

## Methods

### Sample fabrication
The disordered metasurface is fabricated by a combination of electron beam lithography (EBL) and reactive ion etching (RIE) processes. Firstly, a titanium dioxide ($TiO_2$) thin film of 600-nm thick is deposited under electron beam evaporation (EBE) on a smooth glass ($SiO_2$) substrate. Subsequently, a layer of polymethyl methacrylate (PMMA) is spin coated onto the $TiO_2$ thin film. After EBL, development of the PMMA resist is performed. A layer of Cr film is deposited on the PMMA pattern using the EBE process again, followed by washing off the remaining PMMA resist. The $TiO_2$ pattern is then created using RIE process. Finally, the Cr mask is removed by using chemical etching.

### Data acquisition
The camera and the SLM are synchronized via a MATLAB program in data acquisition. The phase-only SLM converts 8-bit grayscale (0–255) to phase delay (0–$2\pi$ in radian). Each phase pattern displayed on the SLM is the summation between a QR code (rescaled to 0–64) and an image from the extracted from the "Large-scale CelebFaces Attributes" (CelebA) dataset (rescaled to 0–64) so that the phase delay can be constrained from 0 to $\pi$ to increase the modulation efficiency. 60,000 human face images are randomly selected from the CelebA and each one is randomly paired with a QR code which represents a number ranging from 0 to 99. Finally, a total of 60,000 combination patterns are sequentially displayed on the SLM. The corresponding speckles are captured and recorded by the camera to paired up as dataset. The combined images (with dimension of $32 \times 32$) loaded on the SLM and the corresponding recorded speckle patterns (with dimension of $256 \times 256$) are paired up for DNN training. Statistic analysis of these 60,000 speckles can be referred to Supplementary Note 9. In this study, 58,000 samples are used for DNN training, and the rest 2000 samples act as the test dataset for DNN evaluation. Notably, samples in the training and test dataset share no overlap.

### DNN training strategy
To recover hidden information (i.e., human faces from CelebA) from the speckle patterns and the authorized key (i.e., the QR code), a deep neural network, namely DMNet, is trained in this study. The structure of DMNet is elaborated in detail in Supplementary Note 3. For network training, the optimization problem is defined in Eq. 4, which includes three terms, the loss function ($L$), the regularization term, and the polarization regulation term. The network output ($\hat{y}$) is generated by inputting the ciphertext ($x_1$) and paired QR code ($x_2$) into the $P(i)$-DMNet ($M_i$) with parameters ($\theta$), i.e., $\hat{y} = M_i(x_1, x_2; \theta)$. The loss function (Eq. 5) consists of a negative PCC (Eq. 6) and a mean square error (MSE, Eq. 7) between the plaintext ($y$) and the network output ($\hat{y}$). The regularization term, the second term in Eq. (4), is the PCC between the plaintext ($y$) and the network output with correct ciphertext ($x_1$) and yet a randomly generated binary-amplitude matrix ($x_{random}$) as inputs, namely the $M(x_1, x_{random}; \theta)$. And the regularization coefficient ($\lambda$) is an empirical parameter that is set to be 0.3 in this study. Notably, the regularization term is to strengthen the dependency of Input 2, so that only correctly paired QR code can guarantee correct decryption. Similarly, the third term of Eq. 4 is the polarization regulation term, enabling the independence among different polarizations. By minimizing the loss function, parameters of the DMNet are optimized via Adam with decoupled weight, namely AdamW, whose learning rate is initially set as 0.005 and decays with cosine annealing for 500 epochs. The training framework is Pytorch 1.9.0 with Python 3.7, using CUDA for GPU acceleration. The computing unit is a Dell precision

workstation with E5-1620v3, 64 Gb RAM and a RTX3090 GPU.

$$\operatorname*{argmin}_{\theta} L\left(y, \hat{y} = M_i(x_1, x_2; \theta)\right) + \lambda \|\mathrm{PCC}(M(x_1, x_{\mathrm{random}}; \theta))\|$$
$$+ \sum_{j, j \neq i} \|\mathrm{PCC}(M_j(x_1, x_2; \theta))\| \tag{4}$$

$$L(y, \hat{y}) = -\mathrm{PCC}(y, \hat{y}) + \mathrm{MSE}(y, \hat{y}) \tag{5}$$

$$\mathrm{PCC}(y, \hat{y}) = \frac{\langle (y - \langle y \rangle)(\hat{y} - \langle \hat{y} \rangle) \rangle}{\sigma_y \sigma_{\hat{y}}} \tag{6}$$

$$\mathrm{MSE}(y, \hat{y}) = \left\langle \|y - \hat{y}\|^2 \right\rangle \tag{7}$$

## Data availability
The data that support the figures and other findings of this study are available from the corresponding authors upon reasonable request.

## Code availability
The code used for the meta-hologram design is available from the corresponding author upon reasonable request. The code of deep neural network used for decryption is available from https://github.com/jian29ye4/double-secure-speckle-encryption. Human face images are available in https://mmlab.ie.cuhk.edu.hk/projects/CelebA.html, which are demonstrated and reproduced with the permission from the author of the dataset.

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

## Acknowledgements

P.L. acknowledges the funding support the National Natural Science Foundation of China (NSFC) (81930048), Hong Kong Innovation and Technology Commission (GHP/043/19SZ, GHP/044/19GD), Hong Kong Research Grant Council (15217721, R5029-19, C7074-21GF), Guangdong Science and Technology Commission (2019BT02X105), Shenzhen Science and Technology Innovation Commission (JCYJ20220818100202005), and Hong Kong Polytechnic University (P0038180, P0039517, P0043485, P0045762). Q.S. acknowledges the funding support from the National Natural Science Foundation of China (12204264), the Shenzhen Science and Technology Innovation Commission (WDZC20220810152404001; JCYJ20230807111706014), and Key Laboratory of Nanodevices and Applications, Suzhou Institute of Nano-Tech and Nano-Bionics, Chinese Academy of Sciences (ZS2304). The authors gratefully acknowledge the use of advanced focused ion beam system (EMO25200) of NSTC 112-2731-M-006-001 and electron beam lithography system belonging to the Core Facility Center of National Cheng Kung University (NCKU). P.C.W. acknowledges the support from the National Science and Technology Council (NSTC), Taiwan (111-2112-M-006-022-MY3; 112-2124-M-006-001), and in part from the Higher Education Sprout Project of the Ministry of Education (MOE) to the Headquarters of University Advancement at NCKU. P.C.W. also acknowledges the Yushan Fellow Program by the MOE, Taiwan for the financial support. The research is also supported in part by Higher Education Sprout Project, Center for Quantum Frontiers of Research & Technology (QFort) at NCKU.

## Author contributions

Z.Y. conceived the idea; Z.Y. and H.L. conducted the numerical computation and experiment and prepared the figures; W.Z. conducted the numerical computation of the disordered metasurface; J.Y., Q.Z. and W.L. assisted in building up the system; P.-S.H., Y.-T.L. and P.C.W. conducted the fabrication; Z.Y., H.L., Q.S., B.L. and P.L. wrote the manuscript. P.G., Q.S. and P.L. supervised the project. All members contributed to the discussion of the results and proofreading of the manuscript.

## Competing interests

The authors declare no competing interests.
