## [Peer Review File · Nature Communications]

High-security Learning-based Optical Encryption assisted by Disordered MetasurfaceREVIEWER COMMENTS

Reviewer #1 (Remarks to the Author):

The manuscript by Yu et al. describes a method using speckle as part of a sophisticated high security encryption method. The speckle is produced by a metasurface. Security results of a combination of a security key attached to the 2D-information being encrypted, and speckle from the metasurface further encrypts the interference between the 2D-information and the key. To retrieve the crypted data, speckle is deciphered using artificial intelligence methods, and more specifically using a deep neural network.

With few exceptions, the English is understandable. The style is clear.

1) General perspective.

1a) The whole combination may well be new, although all its components are known. I have some questions about the whole process (see below, section 2), but publication in an archival journal certainly deserves to be considered.

1b) Concerning the use of a metasurface to produce pseudo-random speckle, I need more explanation to understand why that is appropriate (see below, section 3).

Globally, I would need to understand whether the authors claim that the interest of this manuscript for possible publication in a Nature family journal lies in 1a, and 1b is a side issue, or whether it lies in 1b, and 1a is a side issue, or whether they consider that there is some specific link between 1a and 1b that binds them together (in the latter case, I missed that link). A clear answer to that question is a prerequisite for a positive advice concerning this work.

2) Let me now consider in some more detail the general encryption and deciphering scheme (1a).

What is new in this scheme? How does it compare with reference 24?

The deep neural network (DNN) is described in Supplementary Note 3. Nothing is said about comparing this particular DNN structure with others. Is this choice of structure critical? How much does it outperform others?

If Eqn (10), line 146, is correct, then the deciphering problem is to invert a known matrix. Is the deep neural network really useful here? The issue should at least be discussed.

3) Next, let me address the issue of using a metasurface to produce speckle (1b).

It is said that “conventional scattering media” are instable over time. What kind of conventional scattering media is considered here, and why is it instable? It is said (line 45) that ground glass creates multiply scattered light. I would agree that this is true for other “conventional scattering media” such as milk glass, where roughness is in the whole volume. Such diffusers rely on multiple scattering, but ground glass does not. A ground glass can have a quite low surface roughness, for example 1 or 2 microns, and yet create fully developed speckle by single scattering. The authors’ metasurface diffuser also relies on single scattering.

It is said (line 48) that ground glass diffusers are really deterministic, not random. How about the metasurface diffusers?

I suspect that the “limited memory-effect range” of conventional scattering media (line 75) refers to the thickness effects of thick “conventional scattering media” such as milk glass, which is inherently a consequence of multiple scattering and can be considered as a rather strong random characteristic, much more difficult to unscramble than that of singly scattered light from a ground glass or metasurface. I can understand that singly scattered speckle with a “large memory-effect range” is easier to learn using deep neural networks than multiply scattered speckle, but is that benefic or detrimental to security? An explanation is required here.

It is said (line 351) that hard to predict or irreversible changes occur in conventional scattering media, and quote is made to reference 6 and 23. Is that true for all kinds of conventional scattering media? And if it is a problem with only some of them, why not use those that do not show that detrimental behavior?

Lines 288-318: large noise tests have been conducted with the metamaterial diffuser. Have they also been conducted in the same noise conditions with a conventional ground glass diffuser (a real ground glass, i.e., producing singly scattered speckle). If that is the case, it should be described in the manuscript. Otherwise, I feel that something is missing for a convincing argument against “conventional scattering media”. Line 75 claims that there are “restrictions to practical applications due to inherent defects in existing studies”, therefore you claim that existing studies do not really provide a valuable answer. Tests are missing to substantiate your criticism of “conventional scattering media”.

4) Summary:

To summarize, there is some interesting content in this manuscript. With a convincing answer to my three sets of comments (section 1, 2, 3) I could perhaps recommend publication in Nature Communications. Otherwise, some other publication may be a better choice.

5) Addendum: minor suggestions:

I recommend the authors to consider the following items for a possible improvement of their manuscript as far as the style and the readability are concerned. However, the answer to these items will not affect my recommendation.

- The 2D content to be ciphered and transmitted is called “plaintext”, but it is really an image showing a low-resolution human face from a data base of human faces. Is it customary to call such images “plaintext”? Please cite a reference, as otherwise this denomination may be misleading. Indeed, Figure S4 in the supplementary material suggests that the method may not work as well for text (like the text that you are reading now) as for an image.
- In other words, on lines 58-64: I need to be convinced that a low resolution human face is more “complicated information” than real text.
- Where is the phase profile mentioned in line 32?
- Line 36, multi-channel encryption: how many channels may be encrypted? Polarization is generally a 2-dimensional phenomenon. From the remainder of the manuscript, I would also understand that 2 channels can be encrypted, is that true?
- Line 48, “while multiple scattering can scramble the input information, this seemingly randomness is actually governed by a deterministic transformation defined by the scattering medium”. Are you claiming that metasurface speckle is less deterministic than that?
- Line 70, how do you define a one-to-one mapping between two images?
- Line 147, what do you define “the functions of the plaintext, security, key, and DM”?
- Line 157-158: how does “Bob” know the neural network structure and weights?
- Line 163, you may mean “leak”, not “lack”.
- Line 172 are the phase delays simulated, or are they calculated?
- Line 188, what is “holographic” here? I can see interferences, but not a hologram.
- Line 194, what do you call “spatial security”?

- Line 206, “scale”, not “scalar”.
- Lines 217 and 218, define PCC and SSIM.
- Line 330, what do you mean with “the data transmission channels ... could be numerous”?
- Line 347, are the DM all identical for mass-production? Is this a good security feature?
- Lines 357-359: this sentence is unclear.
- Line 380, do you mean a circularly polarized beam?
- Line 387, what is the advantage of reducing the phase excursion to π instead of 2π ?

Reviewer #2 (Remarks to the Author):

In this manuscript, the authors investigated the use of deep neural networks for decrypting optical encryption via scattering-induced speckle patterns. The authors employed disordered metasurfaces to enhance noise tolerance and memory effect while utilizing a QR-code security key for better security. The geometric phase is also incorporated to further improve security with polarization degrees of freedom.

While this is a solid engineering study, combining well-established prior results in metamaterials, disordered photonics, deep learning photonics, and scattering theory, I do not believe that this work meets the high criteria of Nature Communications.

First, most of the operation principles of the platforms utilized in this manuscript have been intensively studied in previous literature, such as flat optics related to F. Capasso’s works, disorder engineering connected to [24], and the connection of machine learning and scattering [Optics Letters 45, 5279 (2020); Optics and Lasers in Engineering 141 (2021): 106570; OSA Continuum 3.11 (2020): 2968-2975]. Although there is no inherent issue with using well-developed platforms, this approach inevitably falls short in terms of impact if there is insufficient novelty in the other parts.

Second, the use of RCP/LCP degrees of freedom and disorder engineering in this manuscript does not constitute a groundbreaking improvement in device engineering. The polarization degrees of freedom of light are confined to RCP and LCP (or potentially two linear polarizations), which only offers incremental progress. Disorder engineering for the result of

this manuscript is also near the level of [24], showing insufficient impact.

Third, in principle, the relationship between scattering intensity and scatterer distributions is not one-to-one, which should be critical for encryption and decryption. Even a constant intensity profile of waves can be obtained with different scatterer distributions or different incident wave profiles due to the design freedom in the phase evolution inside scattered light. Although such limitation may not be critical in small-size problems addressed in this manuscript, it could be a critical issue in large-scale problems and should be discussed carefully.

Finally, machine learning technique utilized in this work aligns closely with traditional techniques in the field of deep learning photonics: inverse design of materials for optical information. I find insufficient novelty or impact compared to previous literature.

In conclusion, while this study represents solid engineering work, skillfully combining established techniques and platforms for encryption and decryption, it lacks novelty. Thus, it would be more suitable for a specialized journal. Therefore, I cannot recommend this manuscript for acceptance.

Reviewer #3 (Remarks to the Author):

The manuscript titled “High-security Learning-based Optical Encryption assisted by Disordered Metasurface” proposes an optical encryption scheme that makes use of disordered metasurfaces (DM) and neural networks. DM is used to generate speckles and neural networks are employed to establish the relationship between speckles and plaintext. My comments are given as below.

(1) The authors mentioned that DM was utilized to overcome the instability of conventional scattering media (CSM). Instead of DM, SLM can also be used to generate speckles in the optical encryption scheme, which can further enhance the stability. Moreover, SLM can be a dynamic one to generate different patterns for the encryption. Then, I cannot find the indispensable role of DM in this encryption work.

(2) Following the comment above, the basic framework of this work is quite similar to Optics

Express 24, 13738-13743 (2016). The only difference is the authors here used DM to replace one SLM in the OE paper, and separated the whole process into the encryption and decryption processes.

(3) This comment is also about the novelty. Similar optical encryption results have been reported in other works. For example, "Speckle-based optical cryptosystem and its application for human face recognition via deep learning" written by the authors. While the authors here used DM instead of scattering media, the novelty seems insufficient to make it publish in Nature Communications.

(4) There is one more question about the role of DM. The authors used DM to actively manipulate the optical field and designed the encryption process for speckle image generation. In my opinion, the output speckles from DM can be precisely calculated or simulated by commercial software (e.g., CST) from the designed structure of DM and input light. Thus, the training by machine learning for the speckles seems meaningless, because one can obtain the propagation function of DM and then calculate the input information directly from the output speckle.

(5) In general, the plaintext (secret image) should be completely unknown to the decryptor for an image encryption process. Thus, another critical comment is that the plaintext in the encryption process is in a database that is pre-known to both the encryptor (Alice) and the decryptor (Bob). If the plaintext (secret image) is pre-known in a database to Bob, Alice can simply send the order of the secret image, such as a number, to Bob for the decryption, instead of sending a QR code to increase the data amount. In my opinion, this image identification process by Bob cannot be defined as an image decryption process.

(6) Many optical encryption schemes using metasurfaces were published in recent years. The authors should make comparisons with these works to show the advantages of the outlined scheme.

(7) The encryption process, especially the experimental setup, should be covered entirely within the manuscript rather than being included in supporting information.

Reviewer #4 (Remarks to the Author):

The manuscript by Yu et al. presents a new approach of optical encryption by integrating metasurfaces and artificial intelligence. In the encryption process, the designed metasurface

can generate diverse speckle patterns when light modulated by different plaintexts and QR code phase patterns is incident upon it. For the decryption, the receiver can recover the information by feeding the speckle pattern and QR code, i.e., the ciphertext and security key, into a deep neural network. Additionally, polarization multiplexing is further leveraged to enhance the security of the encryption. Overall, the results of this study are clearly demonstrated and may enrich the research on optical encryption with metasurfaces. However, regarding the delivered methodology and result in this manuscript, the following questions and concerns need to be addressed.

1. Recently, various studies of optical encryption with metasurfaces have been reported. Although the authors highlighted this work's unique features compared with previous speckle-based optical encryption, the discussion on its advantage over other metasurface-based encryption schemes is not elaborated. I would suggest the authors include additional discussion on this aspect in the revised manuscript.
2. To the best of my understanding, the receiver (Bob) does not directly utilize the metasurface in the decryption process, and all he required is the ciphertext and security key. It seems that the metasurface is only responsible for mapping the plaintext and security key to the ciphertext in the encryption process, but is not involved in the decryption. Therefore, the function of the metasurface can in principle be replaced by computer simulations. Is this a disadvantage of the proposed approach? Please comment on this point.
3. In Fig.4a, the authors demonstrated the decrypted information when Input 1 is accurate and Input 2 is set to be random binary-amplitude matrices. What would be obtained if Input 1 is accurate while Input 2 is the "correct" QR code for a different plaintext? For example, if we feed the correct Input 1 for plaintext 1 and the correct Input 2 for plaintext 2 into the DMNet, could the information for plaintext 1 or 2 be recognized in the decrypted result?
4. Due to the characteristic of the propagation phase, the polarization-multiplexed metasurface is wavelength-sensitive. Would the generated speckles be different with incident wavelengths other than 488nm? If this difference is very significant, I wonder if it means that different wavelength datasets can be obtained to train more DMNets. In this case, the wavelength information can be further used to improve the security of encryption.
5. According to the Methods, the height and width for the plaintext images/QR codes/speckle patterns are the same, which is however not the case as shown in Fig.3-5. The authors should check carefully to avoid misunderstanding

REVIEWER COMMENTS

Reviewer#1 (Remarks to the Author): *The manuscript by Yu et al. describes a method using speckle as part of a sophisticated high security encryption method. The speckle is produced by a metasurface. Security results of a combination of a security key attached to the 2D-information being encrypted, and speckle from the metasurface further encrypts the interference between the 2D-information and the key. To retrieve the crypted data, speckle is deciphered using artificial intelligence methods, and more specifically using a deep neural network. With few exceptions, the English is understandable. The style is clear.*

1) General perspective. 1a) The whole combination may well be new, although all its components are known. I have some questions about the whole process (see below, section 2), but publication in an archival journal certainly deserves to be considered. 1b) Concerning the use of a metasurface to produce pseudo-random speckle, I need more explanation to understand why that is appropriate (see below, section 3).

Globally, I would need to understand whether the authors claim that the interest of this manuscript for possible publication in a Nature family journal lies in 1a, and 1b is a side issue, or whether it lies in 1b, and 1a is a side issue, or whether they consider that there is some specific link between 1a and 1b that binds them together (in the latter case, I missed that link). A clear answer to that question is a prerequisite for a positive advice concerning this work.

Reply: We sincerely thank the reviewer's critical comment, which could be related to the novelty and motivation of this study. We will try to make it clearer regarding 1a) and 1b). In 1a), a double-secure procedure is proposed to enhance the security to avoid one-to-one mapping between the input and the output in previous studies. A polarization-sensitive disordered metasurface (DM), equipped with combined propagation and geometric phases, is introduced to provide multi-channel polarization

encryption with one DM. In 1b), replacing conventional scattering media (CSM) with DM produces speckles of high-level stability, making it suitable for real-world scenarios such as long-term operation and data extension or updating. The proposed study indeed does not emphasize either 1a) or 1b), but both of them, trying to fulfill one purpose: to build a highly secure speckle-based cryptosystem towards practical applications. Detailed explanation is expanded below.

In 1a), the proposed scheme provides higher-level security than other speckle-based cryptosystems by eliminating the one-to-one mapping between the image and the speckle pattern. As shown in SFig. 1, assuming with an identical setup, previous studies based on speckle encryption can also realize an inverse transformation of the optical system (SFig. 1b, encryption path: input wavefront [plaintext] \rightarrow CSM \rightarrow cipher text [speckles]; decryption path: ciphertext [speckles] \rightarrow plaintext). In contrast, in our realization (SFig. 1a, encryption path: input wavefront [plaintext + QR code] \rightarrow DM \rightarrow ciphertext [speckles]; decryption path: [ciphertext + QR code] \rightarrow plaintext), the encryption path is not a pure inverse of the decryption path. That said, the ciphertext (speckle pattern) contains two components originating from the phase object and the QR code, respectively. Therefore, to decrypt and obtain the plaintext, the network needs to remove the contribution of the QR code from the speckle pattern, so that it can output the pure phase object. That's why in this study, the DMNet is designed with two inputs (speckles and QR code) and one output (plaintext). By doing so, the decryption gains double-secure treatment with both speckles and QR code for correct decryption (Fig. 4a-b: mismatch of the QR code fails to decrypt the image). Hence, the security of the speckle-based cryptosystem is therefore nonlinearly and significantly enhanced.

SFig. 1 Encryption framework comparison between this study (a) and previously reported studies (b). DNN: deep neural network; HWP: half-wave plate; L: lens; P: polarizer; QWP: quarter-wave plate; SLM: spatial light modulator.

As an arbitrary polarization can be decomposed into two orthogonal polarization states (RCP and LCP in this experiment) of different weights, speckles generated from the metasurface, based on the combination of propagation and geometric phases, vary with the polarization of the incident beam. In this work, the information is encrypted in the output speckles instead of relying solely on the metasurfaces themselves. That said, any polarization state could be an encryption channel. The experimental results are shown in SFig. 2, where a combination of a half-wave plate (HWP) and a quarter-wave plate (QWP) (as shown in SFig.2a) is used to alter the polarization of the incident beam. Two specific orthogonal optical channels are defined by two circular polarization states, *i.e.*, P(1): LCP and P(7): RCP. In addition to these two orthogonal channels, five intermediate polarization channels, P(2) to P(6), located between P(1) and P(7), are created by rotating the QWP with an interval of 15° , as shown in the second row in SFig. 2b. Figures in the third row of SFig. 2b show the recorded speckles corresponding

to these seven incident polarizations, and the variation of the Pearson correlation coefficient (PCC) of the speckles taking the speckle of incident LCP as the reference is illustrated in SFig.2c. It can be seen that the speckle is highly sensitive to the rotation angle: the PCC gradually decreases from 1 to 0.08. Such a decrease of PCC can significantly impair the recover efficiency of the input information. From another perspective, the phenomenon also indicates the data independence of each polarization state, which is furtherly verified as shown in S Figs. 2d and 2e, where seven DMNets are individually trained using these seven polarized datasets, and each DMNet trained with P(i) data is denoted as P(i)-DMNet (i=1,2,3,4,5,6,7). With correct QR code (not shown in the SFig. 2 for simplicity), the plaintexts can be correctly deciphered only when the polarization state of the speckle matches that of the corresponding DMNet, as shown in the diagonal in SFig. 2d/e, where P(i) speckles being input into the P(i)-DMNet could results in decryption PCCs of ~0.94. Once the polarization channels between the input data and network are mismatched, *e.g.*, P(1)-speckles (LCP) input into P(7)-DMNet (RCP) or P(7)-speckles (RCP) input into P(1)-DMNet (LCP), the decrypted plaintext exhibits unrecognizable faces, with decryption PCCs of 0.0158 and 0.0268, respectively. In statistical analysis based on SFig. 2e, it can be observed that the decryption PCCs for matched polarization states (~0.94 on the diagonal) are significantly higher than those for mismatched polarizations (<0.06 off the diagonal). That said, realizations for multi-channel decryption do not necessarily rely on the orthogonality of the polarization. The additional polarization states between the orthogonal ones can also support the independence among the polarization channels. By jointly adjusting a half-wave plate and a quarter-wave plate, more polarization states can be introduced. In principle, arbitrary polarization state could be an encryption channel. Therefore, the feasibility of achieving multi-channel encryption, which requires independence of polarization channel and realization of multi-polarization channels based on the DM, is assured.

To make it clearer, related contents have been added to Lines 206-222 and Lines 310-332 in the revision of the manuscript.

SFig. 2 (a) Experimental setup to generate polarization-sensitive speckles in this study. (b) Seven different polarization states between the LCP and RCP are defined by tuning the fast axis of the QWP in the setup, and the resultant speckles corresponding to the 7 different polarization states. (c) The PCC curve of the speckle as a function of polarization state. (d) Cross validation of decryption by inputting speckles of 7 different polarization states (i.e., P(i)-speckles, $i=1,2,3,4,5,6,7$) into the DMNets corresponding to the 7 different states, (i.e., P(i)-DMNet, $i=1,2,3,4,5,6,7$). (e) Averaged decryption PCC corresponding to the cross validation arrangement in (d) and each is averaged over 2,000 samples. DM: disordered metasurface; HWP: half-wave plate; L: lens; PL: polarizer; QWP: quarter-wave plate.

Comment 1b) from the reviewer is about the concern of the contribution from the disordered metasurface (DM) compared with that of conventional scattering media (CSM). We do agree that the function of the DM in this study includes generating

pseudo-random speckles (further discussion in Section 3 below), but it is more than that. Using DM further enables the high-level stability and therefore adding new data without extra network training, which is a breakthrough from the CSM, as discussed in Fig. 5 and the last paragraph in the Discussion Section of the original manuscript. With CSM, like the mentioned work from our own group, the temporal generalization of the trained decryption networks is limited. For example, the decryption function of the network trained by data collected on Day 1 only works well for the validation/testing data collected on Day 1, but fails for data collected on other days, because the external perturbations or tiny changes on the CSM can significantly decorrelate the speckles (e.g., speckle patterns collected on Day 1 can be entirely different from those obtained on other days, even with an identical system). To enhance the generalization capability, the network trained by the Day 1 data has to be further trained by the other days' data. The intervention of DM, however, realizes the temporal generalization by providing featured stability for the speckle generation (Fig. 5). And therefore, even though new information to be encrypted is needed, no additional training for the network trained by the Day 1 data is necessary (Fig. 5). To make it clearer, stability comparison between CSM and DM has been added the revision of the manuscript, and more detailed discussion has been added and refined in the "Polarization channel protection" section in revision.

2) *Let me now consider in some more detail the general encryption and deciphering scheme (1a).*

2.1) *What is new in this scheme? How does it compare with reference 24?*

Reply: We thank the reviewer's insightful comment. The comparison between the captioned work and Reference 24 is explained below.

To begin with, the application scenarios and technical routes in these two works are very different. In Reference 24, the authors demonstrated to achieve fluorescence imaging with large field of view and high numerical aperture (NA) simultaneously. A

numerically calculated pattern displayed on the spatial light modulator (SLM) is used to accurately modulate the wavefront of light and a high NA optical focus can be obtained when the shaped light transmits through the DM, where the DM is conjugated with the SLM accurately in advance with a precise alignment system and procedure and the phase profile of the DM is known as a priori. The introduction of the complex alignment system and the procedure for optical encryption will, however, lead to an encryption system that depends merely on the DM. Such a system, albeit complex, can be easily cracked if the hacker steals the information of the DM. In our work, the DM is used to replace a CSM to mainly overcome the issue of instability, in which the alignment between the input (output) and the DM is not critical. Thus, the effective phase profile (related explanation can be referred to Comment 2.3 and SFig. 3) of the DM is unknown and system dependent. In this scenario, a deep neural network is used to recover the input information from the output speckle. In addition, the combination of the propagation phase and the geometric phase makes the speckle output from the DM sensitive to the polarization of the incident beam, and, in theory, the speckle from any arbitrary polarization channel can be used for encryption (detailed explanation can be referred to the replies to Comment 1). In this sense, the DM acts as a dynamic scattering medium, which greatly enhances the security of the proposed method together with the double-secure procedure.

2.2) The deep neural network (DNN) is described in Supplementary Note 3. Nothing is said about comparing this particular DNN structure with others. Is this choice of structure critical? How much does it outperform others?

Reply: Thanks a lot for the comment. The authors did not give the comparison with other DNN structures here because there are no other studies reporting the same transformation from two inputs (speckles and QR code) to one output (human face), in which the human face and QR code are superposed as the phase object transmitting through the medium, as described in Method. For the reason why the network is

designed in this formation, more details extracted from Supplementary Note 3 are put here for your reference, which tries to match the physical process aiming for better performance: The DMNet composes three parts: 1) Input 2 (the QR code) is first transformed by the complex fully connected layer (Com-FC1); 2) Input 1 (the speckle pattern) concatenated with the transformed Input 2 is fed into the a U-net integrated with densely-connected layers; 3) and the output from 2) is then processed by the other complex fully connected layer (Com-FC2), whose result is the final output of this network. Notably, the use of the complex fully connected layers is to mimic the scattering of the optical field [Nat. Commun., 10: 2029 (2019)], among which the Com-FC1 resembles the forward scattering process and the Com-FC2 does for the inverse scattering process. For the network in 2), it is similar to the DNN [Optica 5: 803-813 (2018)], a U-net based neural network [arXiv:1505.04597, (2015)] integrated with densely connected layers [arXiv:1608.06993, (2016)]. Therefore, this study chooses the network structure that leads to the best performance based on our own practice. In revision, related contents have been added to Lines 403-413 to clarify the concern.

2.3) *If Eqn (10), line 146, is correct, then the deciphering problem is to invert a known matrix. Is the deep neural network really useful here? The issue should at least be discussed.*

Reply: Thanks a lot for the critical comment. As mentioned by the reviewer, we put Eq. (1) on Line 146 here for better discussion:

$$U(x,y,z)=\iint U_P(x_0,y_0)U_S(x_0,y_0)U_{DM}(x_0,y_0)h(x-x_0,y-y_0,z)dx_0dy_0.$$

As seen, the matrices of $U_P(x_0,y_0)$, $U_S(x_0,y_0)$ are known as the plaintext and security key, respectively, which jointly form the input field before the DM. $U_{DM}(x_0,y_0)$, which is determined by the effective phase profile (further explanation can be found in SFig. 3) of the DM, is the function applied onto the input field. The impulse response

$h(x-x_0, y-y_0, z)$ describes the correspondence between the optical field on the detection plane (speckles) and the one out from the DM.

SFig. 3. Illustration of effective phase profile of the DM related to the input in the optical system. A plane wave, modulated by an object to carry the input information with N pixels, transmits through a DM to generate speckles. The area circled by blue dashed lines in the DM is the effective illumination region corresponding to the input. The phase response of this region is the effective phase profile of the DM applied onto the input.

If the pixel-to-pixel alignment between the input and the DM can be tuned accurately with a complex alignment system and procedure, and the alignment between the output and the DM can be tuned accurately using a high NA objective lens to collect all of the scattered light (the exit angle of diffused light from the DM is very large), the input information can be recovered from the output intensity based on some iterative algorithms, such as Gerchberg-Saxton algorithm, with the phase of the DM known a priori in this scenario. However, with increased size for the DM, it is more and more infeasible to collect all scattered light. The introduction of the complex alignment system and procedure will, however, result in an encryption system that solely relies on the DM. Consequently, such a complex system can be easily cracked if the hacker steals the information of the DM. In the meanwhile, the fabrication error of the DM will lead to an inaccurate phase profile, which will further degenerate the input information recovery efficiency, especially for complex input information.

In this work, the correspondence between the DM and the input (output) is arbitrary without an extra alignment system and procedure, and only part of the output field needs to be recorded. Thus, the effective phase profile of the DM is totally unknown and system dependent, resulting in a system-dependent encryption system. Even a hacker might steal the information of the DM, he/she cannot decipher the encrypted information. Under such circumstances, the deep learning method provides a robust solution to learn the correspondence between the input and output regardless of the fabrication error of the DM, system aberration, and misalignment among all optical components. In this work, the effective phase profile of DM is totally unknown due to arbitrary correspondence among optical components in the optical system, but each meta-pillar can be designed independently with a polarization-sensitive structure. Thus, the DM can serve as an optical scattering generator with polarization-sensitive feature.

3) Next, let me address the issue of using a metasurface to produce speckle (1b).
3.1) It is said that “conventional scattering media” are instable over time. What kind of conventional scattering media is considered here, and why is it instable? It is said (line 45) that ground glass creates multiply scattered light. I would agree that this is true for other “conventional scattering media” such as milk glass, where roughness is in the whole volume. Such diffusers rely on multiple scattering, but ground glass does not. A ground glass can have a quite low surface roughness, for example 1 or 2 microns, and yet create fully developed speckle by single scattering. The authors’ metasurface diffuser also relies on single scattering.

Reply: We thank the reviewer for the critical comment. It is agreed that in many optical applications, a CSM like ground glass can be regarded as a stable medium, which, however, is a different story if it is examined in a more micro scale within a longer time window. For a CSM, including ground glass diffuser, the thickness is much larger than the optical wavelength and thus there are three-dimensional optical inhomogeneities. Hence, a conventional disordered medium can be regarded as a multi-layer scattering

medium. Scatterers in each layer distribute randomly and thus the induced optical scattering in each layer is random and spatially different. As a result, the accumulation of optical scattering in each layer is sensitive to environmental perturbations, and the sensitivity increases with the thickness of the scattering medium. In contrast, a DM used in this study is composed of a two-dimensional array of scatterers with uniform height comparable to the optical wavelength, as shown in Fig. 2f in the manuscript. These scatterers are composed of high-refractive-index-contrast, dielectric pillars which act as truncated multimode waveguides. Thus, the DM can be rigorously treated as a single-layer scattering medium with each titanium dioxide meta-pillar fixed in the fused silica substrate, and hence it is less sensitive to environmental perturbations compared with CSM. The stability test (Figs. 5a and c in the main text) also verifies the above comparison. Related comments have also been added to Lines 86-93 and 105-108 in the revised manuscript.

3.2) It is said (line 48) that ground glass diffusers are really deterministic, not random. How about the metasurface diffusers?

Reply: Thanks a lot for raising the point, but the reviewer might misinterpret the meaning of this sentence. We will give more explanation here. For scatterers inside the scattering medium, in a specific time window (which is related the stability of the medium), scatterers distribute randomly but the distribution position of each scatterer is deterministic and unchanged on the spatiotemporal coordinates. Therefore, the transmission matrix of the medium is constant in this specific time window. From the perspective of the output, the output is observed as alternately dark and bright grains (grains are randomly distributed in space), the typical visual feature of speckles, but the intensity distribution is deterministic with the same input, given the same transmission matrix. The DM used in this study also possesses these features but has a much longer scale for the time window to maintain its transmission matrix due to better stability that has been explained in replies to Comment 3.1.

3.3) I suspect that the “limited memory-effect range” of conventional scattering media (line 75) refers to the thickness effects of thick “conventional scattering media” such as milk glass, which is inherently a consequence of multiple scattering and can be considered as a rather strong random characteristic, much more difficult to unscramble than that of singly scattered light from a ground glass or metasurface. I can understand that singly scattered speckle with a “large memory-effect range” is easier to learn using deep neural networks than multiply scattered speckle, but is that benefic or detrimental to security? An explanation is required here.

Reply: We thank the reviewer for the critical comment. The comparison between DM and other CSM (including ground glass) has been discussed in replies to Comments 3.1 and 3.2. In this section, we focus on the relationship between memory-effect range and learning-based recovery efficiency based on the experimental results. The output fields from ground glass and DM are both observed as alternately dark and bright grains, which can be referred to SFig. 4a and b. The average PCCs of recovered images from DM and ground glass by deep learning with the same epochs (500 epochs) are 0.94 and 0.941, respectively, which cannot support the statement of the reviewer. Further, we also attached the training curves for these two media in SFig. 4c, where one cannot see obvious difference between these two curves. In brief, in this deep learning module, there is no clear relationship between the memory-effect range and recovery efficiency. Related comments have been added into the Supplementary Note 7 in the revision.

Sfig. 4 Typical output speckles from a disordered metasurface (DM) (a) and a ground glass diffuser (b). (c) Training loss versus training epoch for data from the DM and the ground glass.

3.4) It is said (line 351) that hard to predict or irreversible changes occur in conventional scattering media, and quote is made to reference 6 and 23. Is that true for all kinds of conventional scattering media? And if it is a problem with only some of them, why not use those that do not show that detrimental behavior?

Reply: We thank the reviewer's critical comment. The comparison between CSM and DM can be referred to the replies to Comment 3.1. In Ref. 24, the authors have compared memory effects of some representative CSMs (ground glass, opal glass and white paint (made of TiO₂ nanoparticles)) with that of DM (shown in Supplementary Figure 2), which can be found in the link at the bottom of this reply. Here we give more explanation and discussion based on our results in combination with their conclusions. DM has a much greater memory effect than that of other media. A ground glass can be regarded as a surface scattering medium with relatively weak scattering in some circumstances, which has the largest memory effect among above three CSMs. However, it is still proved to be an unstable scattering medium in Fig. 5 in our work. In terms of stability, DM is a good choice for speckle-based encryption applications in this study. Moreover, the DM can be mass-producible with mature manufacturing technology, and in the meanwhile it can provide some customized features including polarization and wavelength response.

https://static-content.springer.com/esm/art%3A10.1038%2Fs41566-017-0078-z/MediaObjects/41566_2017_78_MOESM1_ESM.pdf

3.5) Lines 288-318: large noise tests have been conducted with the metamaterial diffuser. Have they also been conducted in the same noise conditions with a conventional ground glass diffuser (a real ground glass, i.e., producing singly scattered speckle). If that is the case, it should be described in the manuscript. Otherwise, I feel

that something is missing for a convincing argument against “conventional scattering media”.

Line 75 claims that there are “restrictions to practical applications due to inherent defects in existing studies”, therefore you claim that existing studies do not really provide a valuable answer. Tests are missing to substantiate your criticism of “conventional scattering media”.

Reply: We thank the reviewer’s insightful comment. The experimental results of ground glass in the same noise conditions were put in Supplementary Note 5 in the previous submission. In the revised version, we have rearranged Fig. 5 and added related results into it, as shown below.

Sfig. 5 (a-b) Stability analysis for the DM-based decryption performance. (a) Background PCC (blue dots) and decryption PCC (red columns) based on the data collected in 14 periods. (b) Decryption performance for three representative examples with respect to the 14 periods in (a). Digits below each reconstructed images are the Decryption PCCs between the decrypted image and the ground truth image. (c-d) Stability analysis for the CSM-based decryption performance. (c) and (d) are the counterparts of (a) and (b), respectively, under the same experiment conditions with a ground glass to replace the DM as the scattering medium.

4) *Summary: To summarize, there is some interesting content in this manuscript. With a convincing answer to my three sets of comments (section 1, 2, 3) I could perhaps recommend publication in Nature Communications. Otherwise, some other publication may be a better choice.*

Reply: We sincerely thank the reviewer for the recognition as well as the insightful comments for us to improve the work. As can be seen, detailed replies have been provided to all comments above and the manuscript has been revised accordingly. Hope the concerns have been addressed or clarified. Nevertheless, if further amendment is needed, please let us know. Thank you.

5) *Addendum: minor suggestions: I recommend the authors to consider the following items for a possible improvement of their manuscript as far as the style and the readability are concerned. However, the answer to these items will not affect my recommendation.*

- *The 2D content to be ciphered and transmitted is called “plaintext”, but it is really an image showing a low-resolution human face from a data base of human faces. Is it customary to call such images “plaintext”? Please cite a reference, as otherwise this denomination may be misleading. Indeed, Figure S4 in the supplementary material suggests that the method may not work as well for text (like the text that you are reading now) as for an image.*

Reply: Please kindly check these two references, Nature Communications 13: 6687 (2022) and Advanced Science 25(9): e2202407 (2022), in which 2D images are the information before encryption and called “plaintext”. Also, definition about “plaintext”, given by Williams, S. (2017). Cryptography and network security: Principles and practices. Ed: Pearson Education, is that “This is the original intelligible message or

data that is fed into the algorithm as input.” Therefore, the information to be encrypted here is not limited to its content (either string, text, or image) and the authors believe that it is proper to call the human face images in this study as “plaintext” since they are the information to be cyphered and transmitted. We have added corresponding references in Ref. 49 in the revised manuscript.

- *In other words, on lines 58-64: I need to be convinced that a low-resolution human face is more “complicated information” than real text.*

Reply: Thanks for this comment. According to the information theory proposed by Shannon [Bell System Technical Journal. 27(3): 379–423 (1948)], higher entropy represents more uncertainty in the information or the image, making the information more complicated. The information complexity can be quantified by the information entropy or Shannon entropy, given by

$$H(I) = - \sum_i P(i) \log_2 P(i),$$

where I is the information (image in this study), $P(i)$ is the probability distribution of grayscale in the image. As calculated, the average entropy for dataset CelebA, fMNIST, quickdraw, and MNIST are 7.57, 3.80, 0.825, and 0.692 bits, respectively, with typical examples provided in Sfig. 6 with their corresponding entropy value (H). As seen, the entropy of human face images is considerably larger than that of clothes, symbols, and digits, possessing much more profound textured and intensity (or grayscale) distribution. Therefore, it is safe to claim that human face images are more complicated information than handwritten digits/texts, as adopted by many other works in the field [Nature Communications 10(1): 2029 (2019); Optica 5(7): 803-813 (2018)]. Related comments have been added to Supplementary Note 4 in revision.

Sfig. 6 Information entropy of some typical examples from four different datasets: CelebA, fMNIST, quickdraw and MNIST.

- Where is the phase profile mentioned in line 32?

Reply: We thank the reviewer's concern and please let us clarify that. The phase profile in Line 32 means the phase or the wavefront of light before it transmits through the DM. Since the incident beam is first modulated by the phase of human face and then the QR code, incident beam carries the phase superposition of the human face and QR code profile as seen in Sfig. 7. And technically, the SLM displays the figure information in Sfig. 7c to modulate the incident beam in experiment.

Sfig. 7 (a) Example of a human face image. (b) Example of a QR code image. (c) The superposition of the human face and QR code information.

- Line 36, multi-channel encryption: how many channels may be encrypted? Polarization is generally a 2-dimensional phenomenon. From the remainder of the manuscript, I would also understand that 2 channels can be encrypted, is that true?

Reply: In the former version of submission, we only demonstrated results of LCP and RCP, which may have caused the concern. As discussed earlier, the output speckles are sensitive to the incident polarization and arbitrary polarization can be used as a channel for encryption. In the revised manuscript, we have added another 5 channels as shown in Fig. 2 and Fig. 5, respectively. Detailed explanation can also be referred to the replies to Comment 1.

- Line 48, “while multiple scattering can scramble the input information, this seemingly randomness is actually governed by a deterministic transformation defined by the scattering medium”. Are you claiming that metasurface speckle is less deterministic than that?

Reply: This comment is related to the explanation of the term “deterministic”. The term “deterministic” can be applied to general scattering media and processes for a specific time window. A disordered metasurface sees no difference in this aspect. Detailed answer can be referred to the replies to Comment 3.2.

- Line 70, how do you define a one-to-one mapping between two images?

Reply: In the previous scheme, the ciphertext (output) is directly used to recover the plaintext(input) in an optical system, and the mapping relationship between ciphertext and plaintext is one-to-one. In this study, the plaintext is first encoded by a security key, and thus the ciphertext is a synthesis of the plaintext and the security key, in which the

mapping relationship between the input (the superposition of plaintext and security key) and output (ciphertext) of the optical system is two-to-one.

- Line 147, what do you define “the functions of the plaintext, security, key, and DM”?

Reply: The phase profiles (amplitude modulation can be also taken into consideration) of plaintext, security key, and the DM cause the corresponding variations of the optical field.

- Line 157-158: how does “Bob” know the neural network structure and weights?

Reply: Alice (Sender) trains data from incident beams of different polarizations in advance and marks each deep neural network with the corresponding polarization. In order to get access to the plaintext, Bob (receiver) must obtain authorization from Alice to acquire the security key and the polarization of the incident beam in advance.

- Line 163, you may mean “leak”, not “lack”.

Reply: The term “lack” means that hackers cannot get the authorization from Alice to acquire the security key and the polarization of the incident beam.

- Line 172 are the phase delays simulated, or are they calculated?

Reply: The phase delays are simulated with commercial software Lumerical FDTD. There is no need to know the detailed phase profile of the DM in experiment, but it only requires that the DM has a randomly distributed phase profile statistically, with some customized features such as polarization sensitivity. Related information has been added the Lines 188 in the revision.

- Line 188, what is “holographic” here? I can see interferences, but not a hologram.

Reply: Thanks for pointing out the typo. The speckle is generated from the interference among light from different modulating elements on the DM. We have deleted “holographic” in the revision.

- Line 194, what do you call “spatial security”?

Reply: Only part of the scattering light field needs to be collected for optical encryption and decryption in this work, and data from different fields of views on the output plane is independent. As a result, spatial security can be introduced.

- Line 206, “scale”, not “scalar”.

Reply: Thanks for pointing out the typo. We have corrected the typo on Line 236 in the updated manuscript.

- Lines 217 and 218, define PCC and SSIM.

Reply: Thanks for pointing out the mistake. PCC has been defined on Line 218 in revision, and the definition of SSIM has been added to Line 262.

- Line 330, what do you mean with “the data transmission channels ... could be numerous”?

Reply: It has been shown that the DM is sensitive to the polarization of incident light, and the output speckle varies with the polarization of the incident beam. Detailed explanation can be referred to the replies to Comment 2.1.

- Line 347, are the DM all identical for mass-production? Is this a good security feature?

Reply: In our work, the efficient phase profile of the DM is system dependent, but the identical features, such as polarization sensitivity, depend on the design principle of each meta-pillar. As the preparation technique is mature, identical features of the DM can be reduplicative. For example, if the DM in operation is impaired or broken, another DM with identical features can be fabricated.

- Lines 357-359: this sentence is unclear.

Reply: As replied to Comment 3.2, the output speckle and the distribution of the scatters are deterministic in a specific time window. As shown in Fig. 5, the speckle output from the DM after several days can still recover the target information, and the performance is close to that in the initial period in a noisy environment; data collected during this period can be directly used to decode the input information without extra training. Further, if the position or orientation of the DM can be adjusted, the status of the medium can be easily recovered back to the initial status, using the PCC of speckles as the reference. Related explanation has been added to Lines 366-388 in the revision.

- Line 380, do you mean a circularly polarized beam?

Reply: We thank the reviewer's careful check of the paper. The answer is yes.

- Line 387, what is the advantage of reducing the phase excursion to π instead of 2π ?

Reply: The adoption was based on past experience. The distributions of pixel values in each image in the dataset are very different. The majority of pixel values in some images concentrate near the maximum value. If the phase excursion is 2π , some effective pixels in these images will be regarded as the background, since the phase value of the background of each image is set to 0. This will cause confusion and affect the reconstruction efficiency via learning.

Reviewer #2 (Remarks to the Author):

In this manuscript, the authors investigated the use of deep neural networks for decrypting optical encryption via scattering-induced speckle patterns. The authors employed disordered metasurfaces to enhance noise tolerance and memory effect while utilizing a QR-code security key for better security. The geometric phase is also incorporated to further improve security with polarization degrees of freedom. While this is a solid engineering study, combining well-established prior results in metamaterials, disordered photonics, deep learning photonics, and scattering theory, I do not believe that this work meets the high criteria of Nature Communications.

First, most of the operation principles of the platforms utilized in this manuscript have been intensively studied in previous literature, such as flat optics related to F. Capasso's works, disorder engineering connected to [24], and the connection of machine learning and scattering [Optics Letters 45, 5279 (2020); Optics and Lasers in Engineering 141 (2021): 106570; OSA Continuum 3.11 (2020): 2968-2975]. Although there is no inherent issue with using well-developed platforms, this approach inevitably falls short in terms of impact if there is insufficient novelty in the other parts.

Reply: We appreciate the reviewer for the critical point about novelty and impact of the work, but we think the concern of the reviewer is some one-sided, which may be due to the way the work was expressed in the previous version. Therefore, we have updated the manuscript with detailed explanations and more experimental results. Regarding the points mentioned above, we would like to give point-by-point reply, as shown below.

1.1 Comparison between our work and Reference 24

To begin with, the application scenarios and technical routes in these two works are very different. In Reference 24, the authors demonstrated to achieve fluorescence imaging with large field of view and high numerical aperture (NA) simultaneously. A

numerically calculated pattern displayed on the spatial light modulator (SLM) is used to accurately modulate the wavefront of light and a high NA optical focus can be obtained when the shaped light transmits through the DM, where the DM is conjugated with the SLM accurately in advance with a precise alignment system and procedure and the phase profile of the DM is known as a priori. The introduction of the complex alignment system and the procedure for optical encryption will, however, lead to an encryption system that depends merely on the DM. Such a system, albeit complex, can be easily cracked if the hacker steals the information of the DM. In our work, the DM is used to replace a CSM to mainly overcome the issue of instability, in which the alignment between the input (output) and the DM is not critical. Thus, the effective phase profile of the DM is unknown and system dependent. In this scenario, a deep neural network is used to recover the input information from the output speckle. In addition, the combination of the propagation phase and the geometric phase makes the speckle output from the DM sensitive to the polarization of the incident beam, and, in theory, the speckle from any arbitrary polarization channel can be used for encryption (detailed explanation can be referred to the replies to Comment 1.2). In this sense, the DM acts as a dynamic scattering medium, which greatly enhances the security of the proposed method together with the double-secure procedure.

1.2 Comparison between our work and other spin-multiplexing metasurface work

Compared with other existing works relying on the geometric phase and propagation phase, the polarization sensitive DM in this work exhibits very different characteristics. In previous works, the information is typically encoded within the metasurface structures, serving as physical carriers that support limited channels (normally two channels with orthogonal polarization states) for effective information transmission. As an arbitrary polarization can be decomposed into two orthogonal polarization states (such as RCP and LCP) of different weights, the output from the metasurface, based on the combination of propagation phase and geometric phase, varies with the polarization

of the incident beam. Therefore, under other incident polarizations apart from the designed polarizations, the transmitted information is invalid due to crosstalk.

In our work, the information is encrypted onto the output speckles instead of relying solely on the metasurfaces themselves, and in the meanwhile these output speckles under different incident polarizations are totally diverse. That is, the optical fields caused by the metasurface are different with different incident polarizations, and any polarization state could be an encryption channel, which can be demonstrated experimentally as below.

The experimental results are shown in SFig. 8, where a combination of a half-wave plate (HWP) and a quarter-wave plate (QWP) (as shown in SFig.8a) is used to alter the polarization of the incident beam. Two specific orthogonal optical channels are defined by two circular polarization states, *i.e.*, P(1): LCP and P(7): RCP. In addition to these two orthogonal channels, five intermediate polarization channels, P(2) to P(6), located between P(1) and P(7), are created by rotating the QWP with an interval of 15° , as shown in the second row in SFig. 8b. Figures in the third row of SFig. 2b show the recorded speckles corresponding to these seven incident polarizations, and the variation of the Pearson correlation coefficient (PCC) of the speckles taking the speckle of incident LCP as the reference is illustrated in SFig.8c. It can be seen that the speckle is highly sensitive to the rotation angle: the PCC gradually decreases from 1 to 0.08. Such a decrease of PCC can significantly impair the recover efficiency of the input information. From another perspective, the phenomenon also indicates the data independence of each polarization state, which is furtherly verified as shown in SFig. 8d and 8e, where seven DMNets are individually trained using these seven polarized datasets, and each DMNet trained with P(i) data is denoted as P(i)-DMNet ($i=1,2,3,4,5,6,7$). With correct QR code (not shown in the SFig. 8 for simplicity), the plaintexts can be correctly deciphered only when the polarization state of the speckle matches that of the corresponding DMNet, as shown in the diagonal in SFig. 8d/e,

where P(i) speckles being input into the P(i)-DMNet could results in decryption PCCs of ~ 0.94 . Once the polarization channels between the input data and network are mismatched, *e.g.*, P(1)-speckles (LCP) input into P(7)-DMNet (RCP) or P(7)-speckles (RCP) input into P(1)-DMNet (LCP), the decrypted plaintext exhibits unrecognizable faces, with decryption PCCs of 0.0158 and 0.0268, respectively. In statistical analysis based on SFig. 8e, it can be observed that the decryption PCCs for matched polarization states (~ 0.94 on the diagonal) are significantly higher than those for mismatched polarizations (< 0.06 off the diagonal). That said, realizations for multi-channel decryption do not necessarily rely on the orthogonality of the polarization. The additional polarization states between the orthogonal ones can also support the independence among the polarization channels. By jointly adjusting a half-wave plate and a quarter-wave plate, more polarization states can be introduced. In principle, arbitrary polarization state could be an encryption channel. Therefore, the feasibility of achieving multi-channel encryption, which requires independence of polarization channel and realization of multi-polarization channels based on the DM, is assured. To make it clearer, related contents have been added to Lines 206-222 and Lines 310-332 in the revision of the manuscript.

SFig. 8 (a) Experimental setup to generate polarization-sensitive speckles in this study. (b) Seven different polarization states between the LCP and RCP are defined by tuning the fast axis of the QWP in the setup, and the resultant speckles corresponding to the 7 different polarization states. (c) The PCC curve of the speckle as a function of polarization state. (d) Cross validation of decryption by inputting speckles of 7 different polarization states (i.e., P(i)-speckles, $i=1,2,3,4,5,6,7$) into the DMNets corresponding to the 7 different states, (i.e., P(i)-DMNet, $i=1,2,3,4,5,6,7$). (e) Averaged decryption PCC corresponding to the cross validation arrangement in (d) and each is averaged over 2,000 samples. DM: disordered metasurface; HWP: half-wave plate; L: lens; PL: polarizer; QWP: quarter-wave plate.

1.3 Comparison between our work and existing works of optical scattering with deep learning

While optical encryption results based on speckles from scattering media have been reported, we would like to clarify that DM here indeed provides more functions in addition to speckle generation. Using DM enables high-level stability and therefore allows for extending the validity of the trained network for new input data obtained on different days without extra network training. This is a meaningful breakthrough from earlier works operating with conventional scattering media, as discussed in Fig. 5 and the last paragraph in the Discussion section of the original manuscript.

With conventional scattering media, like the mentioned work from our own group, the temporal generalization of the trained decryption networks is limited. For example, the decryption function of the network trained by data collected on Day 1 only works well for the validation/testing data collected on Day 1, but fails for data collected on other days, because the external perturbations or tiny changes on the CSM can significantly decorrelate the speckles (e.g., speckle patterns collected on Day 1 can be entirely different from those obtained on other days, even with an identical system). To enhance the generalization capability, the network trained by the Day 1 data has to be further trained by the other days' data. The intervention of DM, however, realizes the temporal generalization by providing featured stability for the speckle generation (Fig. 5). And therefore, even though new information to be encrypted is needed, no additional training for the network trained by the Day 1 data is necessary (Fig. 5).

To make it clearer, stability comparison between CSM and DM has been added to the revision of the manuscript, and more detailed discussion has been added and refined in the “Stability test of the system” section in revision.

SFig. 9 Encryption framework comparison between this study (a) and previously reported studies (b). DNN: deep neural network; HWP: half-wave plate; L: lens; P: polarizer; QWP: quarter-wave plate; SLM: spatial light modulator.

Besides the contribution from the DM, the encryption setting also supports the benefit. As shown in SFig. 9, assuming with an identical setup, previous studies based on speckle encryption can also realize an inverse transformation of the optical system (SFig. 9b, encryption path: input wavefront [plaintext] \rightarrow CSM \rightarrow cipher text [speckles]; decryption path: ciphertext [speckles] \rightarrow plaintext). In contrast, in our realization (SFig. 9a, encryption path: input wavefront [plaintext + QR code] \rightarrow DM \rightarrow ciphertext [speckles]; decryption path: [ciphertext + QR code] \rightarrow plaintext), the encryption path is not a pure inverse of the decryption path. That said, the ciphertext (speckle pattern) contains two components originating from the phase object and the QR code, respectively. Therefore, to decrypt and obtain the plaintext, the network needs to remove the contribution of the QR code from the speckle pattern, so that it can output the pure phase object. That's why in this study, the DMNet is designed with two inputs (speckles and QR code) and one output (plaintext). By doing so, the decryption gains

double-secure treatment with both speckles and QR code for correct decryption (Fig. 4a-b: mismatch of the QR code fails to decrypt the image). Hence, the security of the speckle-based cryptosystem is therefore nonlinearly and significantly enhanced.

Second, the use of RCP/LCP degrees of freedom and disorder engineering in this manuscript does not constitute a groundbreaking improvement in device engineering. The polarization degrees of freedom of light are confined to RCP and LCP (or potentially two linear polarizations), which only offers incremental progress. Disorder engineering for the result of this manuscript is also near the level of [24], showing insufficient impact.

Reply: We appreciate the reviewer for the critical point. As replied to the first comment, the polarization degrees of freedom for speckles are not confined to RCP and LCP. The DM can be used as a dynamic speckle generator by altering the incident polarization. As a result, an arbitrary polarization can be generated and serves as an information transmission channel, and hence this work does not only offer incremental progress in the transmission channel. Related explanation can be referred to replies to Comment 1.2.

Third, in principle, the relationship between scattering intensity and scatterer distributions is not one-to-one, which should be critical for encryption and decryption. Even a constant intensity profile of waves can be obtained with different scatterer distributions or different incident wave profiles due to the design freedom in the phase evolution inside scattered light. Although such limitation may not be critical in small-size problems addressed in this manuscript, it could be a critical issue in large-scale problems and should be discussed carefully.

Reply: We sincerely thank the reviewer for the critical comment. Let us use the classical transmission matrix modal in optical wavefront shaping to address this question. For an

optical system with optical scattering, a transmission matrix (T) can be used to bridge the relationship between the input (E_{in}) and output(E_{out}):

$$E_{out} = T \times E_{in} . \quad (1)$$

For a scattering medium with an infinite boundary, the dimension of T is usually infinite. In real scenarios, the dimension of the measured transmission matrix ($M \times N$) is determined by the dimensions of the input ($M \times 1$) and the output ($N \times 1$). The occasion that a constant output profile of waves can be obtained with different incident wave profiles normally happens when M is larger than N . Let us consider an extreme condition: $M \gg N$, $N=1$, and M is a constant. The output is detected using a single pixel detector and is the interference superposition of all input pixels. In this case, same output can be easily obtained with different input phase profiles using some iterative algorithms like genetic algorithm. With the increase of the number of pixels in the output, such a condition is harder to be met as the rank of T becomes larger. In this experiment, $M=1024$ and $N=65536$. We have also provided the statistical histogram about the similarity of speckles from 60,000 human face phase images as below, in which the probability density for similarity to be larger than 0.9 approaches zero, as discussed in Supplementary Note 9 in the revision.

SFig. 10 Probability density function of mutual PCC among speckles corresponding to 60,000 human face images. The mutual PCC is calculated based on each pair of speckle patterns generated from different face images.

Finally, machine learning technique utilized in this work aligns closely with traditional techniques in the field of deep learning photonics: inverse design of materials for optical information. I find insufficient novelty or impact compared to previous literature.

Reply: We appreciate the reviewer for the critical point. The difference between our work and the existing works in terms of the design and function of the neural network has been discussed in replies to Comment 1.

In conclusion, while this study represents solid engineering work, skillfully combining established techniques and platforms for encryption and decryption, it lacks novelty. Thus, it would be more suitable for a specialized journal. Therefore, I cannot recommend this manuscript for acceptance.

Reply: We sincerely thank the reviewer for the recognition as well as the critical comments for us to improve the work. As seen, detailed replies have been provided to all comments above and the manuscript has been revised accordingly. Hope the reviewer's concerns, especially regarding the novelty of the work, have been addressed or clarified. Nevertheless, if further amendment is needed, please let us know. Thank you.

Reviewer #3 (Remarks to the Author): *The manuscript titled “High-security Learning-based Optical Encryption assisted by Disordered Metasurface” proposes an optical encryption scheme that makes use of disordered metasurfaces (DM) and neural networks. DM is used to generate speckles and neural networks are employed to establish the relationship between speckles and plaintext. My comments are given as below.*

(1) The authors mentioned that DM was utilized to overcome the instability of conventional scattering media (CSM). Instead of DM, SLM can also be used to generate speckles in the optical encryption scheme, which can further enhance the stability. Moreover, SLM can be a dynamic one to generate different patterns for the encryption. Then, I cannot find the indispensable role of DM in this encryption work.

Reply: We sincerely thank the reviewer for raising the critical point. It is true that an SLM can be used to generate speckles with a random phase displayed on the SLM. In the meanwhile, we do agree that the SLM can also enhance the stability since its pixel size is relatively large ($\sim 6 \mu\text{m}$), which can significantly tolerate the environmental vibrations applied to the system. For metasurface, to enlarge the effective size of minimum phase modulation unit, neighboring meta-pillars (e.g., with formation of 2x2, 3x3, 4x4 array, $\sim 0.3 \mu\text{m}$ for each pillar) can be designed with the same phase delay. By doing so, the DM-based system can be further enhanced in principle. Yet, based on our experimental results as shown in Fig. 5 of the revised manuscript, the DM-based system has shown featured stability over the CSM-based system. The stability is thus sufficient for the purpose of optical encryption due to the single layer structure of the DM, ensuring excellent temporal generalization of the encryption system and avoiding additional training for the networks when new data is added. Moreover, the usage of DM in this work has multiple purposes in addition to speckle generation and stability enhancement. For example, DM can support dynamic manipulation in a passive and compact manner, which will be discussed in detail below. Note that the SLM shown in

Fig. 3 in the revised manuscript is used to load the superposed phase patterns of the target with the QR code, but not to generate speckles.

As any polarization can be decomposed into two orthogonal polarization states (RCP and LCP in this experiment) with different weights, speckles generated from the DM, based on the combination of propagation phase and geometric phase, vary with the polarization of the incident beam. In this work, the information is encrypted onto the output speckles instead of relying solely on metasurfaces themselves. Thus, any polarization state could be an independent encryption channel. The experimental results are shown in SFig. 11, where a combination of a half-wave plate (HWP) and a quarter-wave plate (QWP) (as shown in SFig. 11a) is used to alter the polarization of the incident beam. Two specific orthogonal optical channels are defined by two circular polarization states, *i.e.*, P(1): LCP and P(7): RCP. In addition to these two orthogonal channels, five intermediate polarization channels, P(2) to P(6), located between P(1) and P(7), are created by rotating the QWP with an interval of 15° , as shown in the second row in SFig. 11b. Figures in the third row of SFig. 2b show the recorded speckles corresponding to these seven incident polarizations, and the variation of the Pearson correlation coefficient (PCC) of the speckles taking the speckle of incident LCP as the reference is illustrated in SFig. 11c. It can be seen that the speckle is highly sensitive to the rotation angle: the PCC gradually decreases from 1 to 0.08. Such a decrease of PCC can significantly impair the recover efficiency of the input information. From another perspective, the phenomenon also indicates the data independence of each polarization state, which is furtherly verified as shown in SFig. 8d and 8e, where seven DMNets are individually trained using these seven polarized datasets, and each DMNet trained with P(i) data is denoted as P(i)-DMNet ($i=1,2,3,4,5,6,7$). With correct QR code (not shown in the SFig. 11 for simplicity), the plaintexts can be correctly deciphered only when the polarization state of the speckle matches that of the corresponding DMNet, as shown in the diagonal in SFig. 11d/e, where P(i) speckles being input into the P(i)-DMNet could results in decryption PCCs of ~ 0.94 . Once the polarization channels

between the input data and network are mismatched, *e.g.*, P(1)-speckles (LCP) input into P(7)-DMNet (RCP) or P(7)-speckles (RCP) input into P(1)-DMNet (LCP), the decrypted plaintext exhibits unrecognizable faces, with decryption PCCs of 0.0158 and 0.0268, respectively. In statistical analysis based on SFig. 11e, it can be observed that the decryption PCCs for matched polarization states (~ 0.94 on the diagonal) are significantly higher than those for mismatched polarizations (< 0.06 off the diagonal). That said, realizations for multi-channel decryption do not necessarily rely on the orthogonality of the polarization. The additional polarization states between the orthogonal ones can also support the independence among the polarization channels. By jointly adjusting a half-wave plate and a quarter-wave plate, more polarization states can be introduced. In principle, arbitrary polarization state could be an encryption channel. Therefore, the feasibility of achieving multi-channel encryption, which requires independence of polarization channel and realization of multi-polarization channels based on the DM, is assured.

To make it clearer, related contents have been added to Lines 206-222 and Lines 310-332 in the revision of the manuscript.

The compact size of DM makes it easy to integrate with other system components: the size of individual pixel of the DM is $350 \text{ nm} \times 350 \text{ nm}$ in this experiment, which is about $1/400$ of that of a regular SLM (typically around $6500 \text{ nm} \times 6500 \text{ nm}$ or above). In order to operate the SLM, a controller of large size ($\sim 1 \text{ cm} \times 3 \text{ cm} \times 10 \text{ cm}$) is also indispensable. For example, the DM can be integrated with a CMOS chip to constitute a very compact scattering system for optical encryption [Nanophotonics, 9(10): 3071-3087 (2020)]. In the meanwhile, some other issues and drawbacks like instability induced by heating under continuous power supply, calibration for different wavelength ranges, unstable performance under long-term operation, and purchase cost associated with the usage of SLM as a scattering generator cannot be ignored. Note that the SLM

in our system is only used to generate phase patterns, which can be easily replaced by some passive components such as phase test targets.

Fig. 11 Experimental setup to generate polarization-sensitive speckles in this study. (b) Seven different polarization states between the LCP and RCP are defined by tuning the fast axis of the QWP in the setup, and the resultant speckles corresponding to the 7 different polarization states. (c) The PCC curve of the speckle as a function of polarization state. (d) Cross validation of decryption by inputting speckles of 7 different polarization states (i.e., P(i)-speckles, $i=1,2,3,4,5,6,7$) into the DMNets corresponding to the 7 different states, (i.e., P(i)-DMNet, $i=1,2,3,4,5,6,7$). (e) Averaged decryption PCC corresponding to the cross validation arrangement in (d) and each is averaged over 2,000 samples. DM: disordered metasurface; HWP: half-wave plate; L: lens; PL: polarizer; QWP: quarter-wave plate.

In addition, DM could be used to support manipulation of light with multiple degrees of freedom. The combination of geometric phase and propagation phase makes DM sensitive to the polarization of the incident beam. In the meanwhile, the utilization of propagation phase makes each submicron-scale meta-pillar inherently sensitive to the wavelength of incident light. Screening out structures with diverse slopes of phase vs. wavelength curves enables wavelength-sensitive DM [Nano Lett., 18, 12, 8016–8024 (2018)]. In contrast, screening out structures with similar slopes of phase vs. wavelength curves enables broadband feature [Nano Lett, 20, 4, 2791–2798 (2020)]. Such flexibility is not supported if an SLM is used to generate speckles. Last but not least, the concept based on the usage of DM can be extended for a wide range of electromagnetic spectrum from ultraviolet to terahertz light, provided with proper choice of low-loss materials for the meta-atoms. Related contents have been added to Lines 460-462 in the revision.

(2) Following the comment above, the basic framework of this work is quite similar to Optics Express 24, 13738-13743 (2016). The only difference is the authors here used DM to replace one SLM in the OE paper and separated the whole process into the encryption and decryption processes.

Reply: The authors appreciate a lot for the reviewer providing the captioned paper for our reference. In that paper, two SLMs are indeed involved, which, however, do not act as a disorder medium (while the DM in our study acts as a disorder medium) but only as a display to display the phase objects. Specifically, one object image is displayed on both SLMs to enhance the performance. Related description is copied from the second paragraph of Section 3 in that paper for your reference, Experimental demonstration: “... The central 50×50 pixel region of images in the LFW database was clipped and was displayed in magnified form on the central 500×500 pixel region of both SLMs. To reduce the learning cost, SLM1 and SLM2 displayed the same image for capturing a speckle pattern...”. Therefore, the functions of the DM in our study (as detailed in

replies to Comment 1) and that of the SLM in the mentioned study are entirely different. Hence, we believe that the comparison between them should be out of scope.

(3) This comment is also about the novelty. Similar optical encryption results have been reported in other works. For example, “Speckle-based optical cryptosystem and its application for human face recognition via deep learning” written by the authors. While the authors here used DM instead of scattering media, the novelty seems insufficient to make it publish in Nature Communications.

Reply: Thanks for this critical comment. While optical encryption results based on speckles with scattering media have been reported, including the mentioned work from our own group, the authors would like to highlight that DM here indeed provides more functions other than speckle generation only, as detailed in replies to Comment 1.

The mentioned paper presents a proof-of-concept study by encrypting the information by optical speckles based on a ground glass diffuser, whose sole function is to generate optical speckles. Two inherent drawbacks significantly limit its practical applications and its security. First, the diffuser induces multiple scattering (narrow memory effect) where speckles arise so that the speckle pattern or its spatial distribution of grains is susceptible to the external perturbations to the system and/or the diffuser itself. Such instability makes the trained network fail to decrypt the information in even one hour, and hence additional training with new training data is inevitable to adapt to the new medium status. Second, the decryption scheme is one-to-one mapping (outputs plaintext with speckles as input) with one single channel, which is much less secure than the proposed study. To be more specific, the authors would like to underline more about the novelty of the current study.

Using DM further enables the high-level stability and therefore adding new data without extra network training, which is a breakthrough from the CSM, as discussed in

Fig. 5 and the last paragraph in the Discussion Section of the original manuscript. With CSM, like the mentioned work from our own group, the temporal generalization of the trained decryption networks is limited. For example, the decryption function of the network trained by data collected on Day 1 only works well for the validation/testing data collected on Day 1, but fails for data collected on other days, because the external perturbations or tiny changes on the CSM can significantly decorrelate the speckles (e.g., speckle patterns collected on Day 1 can be entirely different from those obtained on other days, even with an identical system). To enhance the generalization capability, the network trained by the Day 1 data has to be further trained by the other days' data. The intervention of DM, however, realizes the temporal generalization by providing featured stability for the speckle generation (Fig. 5). And therefore, even though new information to be encrypted is needed, no additional training for the network trained by the Day 1 data is necessary (Fig. 5). To make it clearer, stability comparison between CSM and DM has been added to the revised manuscript, and more detailed discussion has been added and refined in the "Polarization channel protection" section in revision.

SFig. 12 Encryption framework comparison between this study (a) and previously reported studies (b). DNN: deep neural network; HWP: half-wave plate; L: lens; P: polarizer; QWP: quarter-wave plate; SLM: spatial light modulator.

As shown in SFig. 12, assuming with an identical setup, previous studies based on speckle encryption can also realize an inverse transformation of the optical system (SFig. 12b, encryption path: input wavefront [plaintext] \rightarrow CSM \rightarrow cipher text [speckles]; decryption path: ciphertext [speckles] \rightarrow plaintext). In contrast, in our realization (SFig. 12a, encryption path: input wavefront [plaintext + QR code] \rightarrow DM \rightarrow ciphertext [speckles]; decryption path: [ciphertext + QR code] \rightarrow plaintext), the encryption path is not a pure inverse of the decryption path. That said, the ciphertext (speckle pattern) contains two components originating from the phase object and the QR code, respectively. Therefore, to decrypt and obtain the plaintext, the network needs to remove the contribution of the QR code from the speckle pattern, so that it can output the pure phase object. That's why in this study, the DMNet is designed with two inputs (speckles and QR code) and one output (plaintext). By doing so, the decryption gains double-secure treatment with both speckles and QR code for correct decryption (Fig. 4a-b: mismatch of the QR code fails to decrypt the image). Hence, the security of the speckle-based cryptosystem is therefore nonlinearly and significantly enhanced.

(4) There is one more question about the role of DM. The authors used DM to actively manipulate the optical field and designed the encryption process for speckle image generation. In my opinion, the output speckles from DM can be precisely calculated or simulated by commercial software (e.g., CST) from the designed structure of DM and input light. Thus, the training by machine learning for the speckles seems meaningless, because one can obtain the propagation function of DM and then calculate the input information directly from the output speckle.

Reply: We thank the reviewer’s critical comment. The detailed answer to this comment is expanded below in terms of the role of DM and the function of deep learning in the system.

As mentioned by the reviewer, we put Eq. (1) in the main text here for better discussion:

$$U(x,y,z)=\iint U_P(x_0,y_0)U_S(x_0,y_0)U_{DM}(x_0,y_0)h(x-x_0,y-y_0,z)dx_0dy_0.$$

As seen, the matrices of $U_P(x_0,y_0)$, $U_S(x_0,y_0)$ are known as the plaintext and security key, respectively, which jointly form the input field before the DM. $U_{DM}(x_0,y_0)$, which is determined by the effective phase profile (further explanation can be found in SFig. 3) of the DM, is the function applied onto the input field. The impulse response $h(x-x_0,y-y_0,z)$ describes the correspondence between the optical field on the detection plane (speckles) and the one out from the DM.

SFig. 13. Illustration of effective phase profile of the DM related to the input in the optical system. A plane wave, modulated by an object to carry the input information with N pixels, transmits through a DM to generate speckles. The area circled by blue dashed lines in the DM is the effective illumination region corresponding to the input. The phase response of this region is the effective phase profile of the DM applied onto the input.

If the pixel-to-pixel alignment between the input and the DM can be tuned accurately with a complex alignment system and procedure, and the alignment between the output and the DM can be tuned accurately using a high NA objective lens to collect all of the scattered light (the exit angle of diffused light from the DM is very large), the input

information can be recovered from the output intensity based on some iterative algorithms, such as Gerchberg-Saxton algorithm, with the phase of the DM known a priori in this scenario. However, with increased size for the DM, it is more and more infeasible to collect all scattered light. The introduction of the complex alignment system and procedure will, however, result in an encryption system that solely relies on the DM. Consequently, such a complex system can be easily cracked if the hacker steals the information of the DM. In the meanwhile, the fabrication error of the DM will lead to an inaccurate phase profile, which will further degenerate the input information recovery efficiency, especially for complex input information.

In this work, the correspondence between the DM and the input (output) is arbitrary without an extra alignment system and procedure, and only part of the output field needs to be recorded. Thus, the effective phase profile of the DM is totally unknown and system dependent, resulting in a system-dependent encryption system. Even a hacker might steal the information of the DM, he/she cannot decipher the encrypted information. Under such circumstances, the deep learning method provides a robust solution to learn the correspondence between the input and output regardless of the fabrication error of the DM, system aberration, and misalignment among all optical components. In this work, the effective phase profile of DM is totally unknown due to arbitrary correspondence among optical components in the optical system, but each meta-pillar can be designed independently with a polarization-sensitive structure. Thus, the DM can serve as an optical scattering generator with polarization-sensitive feature.

(5) In general, the plaintext (secret image) should be completely unknown to the decryptor for an image encryption process. Thus, another critical comment is that the plaintext in the encryption process is in a database that is pre-known to both the encryptor (Alice) and the decryptor (Bob). If the plaintext (secret image) is pre-known in a database to Bob, Alice can simply send the order of the secret image, such as a number, to Bob for the decryption, instead of sending a QR code to increase the data

amount. In my opinion, this image identification process by Bob cannot be defined as an image decryption process.

Reply: We thank the reviewer for the critical comment. We will try to clarify the proposed scheme from two aspects.

- (1) The database for training the decryption network, i.e., DMNet, is indeed unknown to both Alice and Bob, and they principally do not need to know the database. The training data can be immediately deleted once the training for the DMNet is completed, so that the risk of data leakage can be mostly minimized. Also, the functionality of the trained DMNets is not driven by the database but just needs the same input of plaintexts with one QR code, which does not increase the data amount. Sending the order of secret images for identification as mentioned by the reviewer, however, greatly depends on the database and, when more information needs to be encrypted, the data amount will considerably increase, which inevitably is at high risk of data leakage. In this regard, it would be a good choice to find an encryption method (independent on the database), which is one of the highlights of this study.
- (2) Regarding the encryption process, to encrypt the plaintext, all that Alice needs to do (or by a third party like the system supplier) is to load the plaintext with a QR code to the encryption system and then obtain the ciphertext. Alice therefore does not need to know any knowledge of the database. For the decryption process, Bob merely needs to put the QR code with the ciphertext into the DMNet with correct weights and then the plaintext comes out, since the DMNets have been trained to do so. Also, before the weights of DMNets (as one key) are delivered to the users, the weights have been effectively optimized by the supplier; the database and the process to train and optimize the DMNets are not open to users, as discussed in (1).

Related contents have been added to Lines 63-66 in the revision.

(6) Many optical encryption schemes using metasurfaces were published in recent years. The authors should make comparisons with these works to show the advantages of the outlined scheme.

Reply: This is a very good suggestion. In almost all of existing studies of optical encryption with metasurfaces, the information is encrypted into the metasurface structures and use them as physical carriers, which can be mainly divided into three categories: (1) single/multi degrees of freedom of light modulation (wavelength, polarization, nonlinear effect, incident angle and OAM); (2) tunable modulation with external stimulus such as chemical reactions, surrounding medium variation, mechanical actuation, electrical gating, and phase transition; (3) other specific mechanisms such as computational ghost imaging with single-pixel imaging (SPI), code division multiplexing, and secret sharing with two aligned metasurfaces. The strategies in these works limit information sharing and compatibility with digital information processing technology. In our work, information is encrypted into output speckles other than metasurfaces themselves. In the meanwhile, the DM can support multi-channel encryption, greatly increasing information capacity and compatibility with digital information processing technologies. Detailed comparisons have been added in the Discussion section (Lines 419-432) in the revision.

(7) The encryption process, especially the experimental setup, should be covered entirely within the manuscript rather than being included in supporting information.

Reply: We sincerely thank the reviewer for the critical suggestion. We have added the experimental setup into Fig. 3 in the updated manuscript, which is also attached below (SFig. 14).

SFig. 14 Decryption performance based on DMNet. (a) The schematic diagram of the optical setup. (b) Examples of plaintext for encryption. (c) The corresponding ciphertexts, *i.e.*, the speckles. (d) Exemplified QR codes. (e) The decrypted information by inputting (c) and (d) into the DMNet. The DMNet herein is trained by the RCP data. Inset numbers below each image in (d) are formatted as PCC(SSIM) between (b) the ground truth and (e) the decrypted images. Abbreviations: DM: disordered metasurface; HWP: half-wave plate; PCC: Pearson's correlation coefficient; RCP: Right-handed circular polarization; QR: quick response; QWP: quarter-wave plate; SSIM: structure similarity.

Reviewer #4 (Remarks to the Author):

The manuscript by Yu et al. presents a new approach of optical encryption by integrating metasurfaces and artificial intelligence. In the encryption process, the designed metasurface can generate diverse speckle patterns when light modulated by different plaintexts and QR code phase patterns is incident upon it. For the decryption, the receiver can recover the information by feeding the speckle pattern and QR code, i.e., the ciphertext and security key, into a deep neural network. Additionally, polarization multiplexing is further leveraged to enhance the security of the encryption. Overall, the results of this study are clearly demonstrated and may enrich the research on optical encryption with metasurfaces. However, regarding the delivered methodology and result in this manuscript, the following questions and concerns need to be addressed.

1. Recently, various studies of optical encryption with metasurfaces have been reported. Although the authors highlighted this work's unique features compared with previous speckle-based optical encryption, the discussion on its advantage over other metasurface-based encryption schemes is not elaborated. I would suggest the authors include additional discussion on this aspect in the revised manuscript.

Reply: We thank the reviewer for the recognition and the insightful comments. In almost all of existing studies of optical encryption with metasurfaces, the information is encrypted into the metasurface structures and use them as physical carriers, which can be mainly divided into three categories: (1) single/multi degrees of freedom of light modulation (wavelength, polarization, nonlinear effect, incident angle and OAM); (2) tunable modulation with external stimulus such as chemical reactions, surrounding medium variation, mechanical actuation, electrical gating, and phase transition; (3) other specific mechanisms such as computational ghost imaging with single-pixel imaging (SPI), code division multiplexing, and secret sharing with two aligned metasurfaces. The strategies in these works limit information sharing and compatibility

with digital information processing technology. In our work, information is encrypted into output speckles other than metasurfaces themselves. In the meanwhile, the DM can support multi-channel encryption, greatly increasing information capacity and compatibility with digital information processing technologies. Detailed comparisons have been added in the Discussion section (Lines 419-432) in the revision.

2. To the best of my understanding, the receiver (Bob) does not directly utilize the metasurface in the decryption process, and all he required is the ciphertext and security key. It seems that the metasurface is only responsible for mapping the plaintext and security key to the ciphertext in the encryption process, but is not involved in the decryption. Therefore, the function of the metasurface can in principle be replaced by computer simulations. Is this a disadvantage of the proposed approach? Please comment on this point.

Reply: Thanks a lot for this comment, which could involve a more general comparison between the software and hardware encryption. Computer simulations or software-based encryption can achieve many functions with no doubt but, in this study, hardware-based encryption could be a better choice. Three major benefits could be gained from this proposed optical cryptosystem.

1) **Security:** With computer simulation, the encryption process is digital since it relies on operation systems, which can be easily invaded and infiltrated by hackers. The whole encryption algorithm/flow or the transformation matrix (or related data) that have to be saved on hard disk can be duplicated by hackers who may technically know the whole encryption process. This could significantly lower down the difficulty to crack the system. Also, the simulations could be modified or taken over by hackers, which may disrupt or disable the entire cryptosystem. Therefore, this study prefers hardware encryption, which allows the whole encryption system to occur on an isolated device/system. The physical encryption process will be

digitally inaccessible and hard to be duplicated (more detailed discussion is given in the next paragraph), even if an attacker knows the phase profile of the metasurface. By doing so, even if the attacker has access to the encryption data (as the data needs hard disk for storage or network for transmission), he will fail to decrypt or understand the data. Besides the protection for the encryption flow, the security level can also be enhanced by introducing the physical properties in more additional dimensions. A hardware-based cryptosystem with a customized metasurface enables more complicated encryption such as encoding information with different optical parameters (like wavelength, polarization and even quantum state). The proposed approach, based on hardware system, can therefore include more physical properties for higher-level security, which might not be easily realized by computer simulations.

- 2) Duplication: The proposed method encrypts information based on an optical system. To duplicate the encryption system, the optical system, especially with all the fine alignments and conjugations, should be accurately re-presented. Since the cyphertext, i.e., speckle pattern in this study, is very sensitive to the settings (such as wavelength, polarization, beam shape, k-vector, and illumination/detection area, etc.) in the optical system, and no strict optical conjugations and/or alignment are needed to induce speckles. Such diversity indeed adds one more dimension to protect the system from being duplicated by others without knowing the physical hardware system details. In contrast, for computer-based simulations, the encryption system duplication can be easily done via a single click.
- 3) Processing speed: Trade-off between the encryption speed and the key length (e.g., complexity of the DM in this study) can be avoided in this study. Since the proposed encryption is based on optical system, the encryption speed is indeed the speed of light, which is irrelevant to the medium (either clear medium or disordered medium). That said, even with a much more complex or large-scale design, the encryption still runs at the speed of light, and the processing time will not be of concern.

3. In Fig.4a, the authors demonstrated the decrypted information when Input 1 is accurate and Input 2 is set to be random binary-amplitude matrices. What would be obtained if Input 1 is accurate while Input 2 is the “correct” QR code for a different plaintext? For example, if we feed the correct Input 1 for plaintext 1 and the correct Input 2 for plaintext 2 into the DMNet, could the information for plaintext 1 or 2 be recognized in the decrypted result?

Reply: We thank the reviewer’s insightful comment, which can further support the high security provided by the proposed method. Related results can be seen in SFig. 15. When the input speckle (Input 1) and QR code (Input 2) are matched, high-fidelity outputs can be obtained (the images on the second row). But if Input 1 is kept unchanged while the QR code (denoted as “Mismatched Input 2”) are swapped with the one corresponding to other samples, the network output (denoted as “Mismatched output”) fails to recover the human faces, yet with similar patterns as shown in the main text (Fig. 4a). Related observations including the figure and description have been added to Supplementary Note 6 in the revision.

SFig. 15 Decryption results with matched and unmatched input pairs. For the unmatched set, Input 1 (speckle pattern) is kept correct but the QR code is switched with its counterpart corresponding to other samples.

4. Due to the characteristic of the propagation phase, the polarization-multiplexed metasurface is wavelength-sensitive. Would the generated speckles be different with incident wavelengths other than 488nm? If this difference is very significant, I wonder if it means that different wavelength datasets can be obtained to train more DMNets. In this case, the wavelength information can be further used to improve the security of encryption.

Reply: This is a very insightful comment. Different wavelength datasets can be obtained to train more DMNets if this difference is significant. Detailed explanations and measures are shown below. The propagation phase of meta-pillar with height of 600 nm is sensitive to the wavelength of light, which can be referred to SFig. 16a. In order to generate wavelength-sensitive speckles, meta-pillars with diverse correspondence between the propagation phase and wavelength are screened out. Such a procedure allows the wavelength to serve as another encryption parameter, in which more channels in another dimension can support independent information transmission and thus more DMNets can be trained for these data. To further improve the sensitivity, meta-pillars with greater heights (e.g., 1000 nm) can be considered, as simulated in SFig. 16b. As shown, the curve of pillar with height of 1000 nm has a more complex correspondence between the propagation phase and the wavelength compared with its peer with a height of 600 nm. This suggests a more wavelength-sensitive feature. Related contents have been added to Supplementary Note 8 in the revision.

SFig. 16 The simulated curves of propagation phase vs. wavelength for meta-pillars with heights of 600 nm (a) and 1000 nm (b). Periodic constant (P) of 350 nm, and the lengths of two axis (u and v) of meta-pillars are 260 nm and 125 nm.

5. According to the Methods, the height and width for the plaintext images/QR codes/speckle patterns are the same, which is however not the case as shown in Fig.3-5. The authors should check carefully to avoid misunderstanding.

Reply: We thank the reviewer for the careful observation. We have carefully tuned all plaintext images/QR codes/speckle patterns into square shapes in the revised submission. Greatest thanks to you for the constructive comments that have helped us improve the expression of the work.

REVIEWERS' COMMENTS:

Reviewer #1 (Remarks to the Author):

I acknowledge that the authors have provided a genuine revision of their initial manuscript together with an impressive amount of detail in a rebuttal document addressing the comments by all reviewers.

While I did write in my initial review that the publication of this work in an archival journal deserves to be considered, I am sorry to say that the whole revision work did not convince me. The rebuttal document contains many words in reply to my questions and criticism, but almost all are just repetitions of many aspects of the initial manuscript content that I had already understood. My questions have not really been addressed. My conclusion is that this work does not meet the quality standards for publication in a Nature group publication. For a brief explanation about my recommendation, I would like to highlight two points, omitting deep learning techniques.

The authors and I agree that there is nothing new about encryption methods in their work. The fact of adding complementary data to the data being encrypted (“plain text”) and merging the two in some complex operation, so that a key containing the complementary data needs to be known by the addressee is a standard technique in encryption. The fundamental contribution of this work does not lie in encryption methods.

The authors repeatedly claim that the disordered metasurfaces are more stable in time than conventional scattering materials. They also claim that they have experimentally tested that claim but they do not provide a comprehensive description of what they did, and whether it is convincing proof that the two samples (DM and CSM) were comparable in a meaningful way. If that is true, it deserves publication, maybe in a Nature group publication, for its own sake. That a thick diffuser providing strong mixing of the “plain text” data (just like added key data, in fact) is more sensitive to minute motions or environmental changes is not a question.

To conclude, I am sorry that I do not recommend the revised manuscript for publication in Nature Communications.

Reviewer #2 (Remarks to the Author):

In this resubmission, the authors provided a thorough revision in response to the reviewers' comments. The revised manuscript highlights the following critical points. First, the authors showed that the suggested platforms operate not merely with two-channel orthogonal polarization but with their superposition, resulting in an enhanced information capacity. Second, the authors established the validity of their method to a specific yet broad class of problems, skillfully evading an issue related to one-to-many relationship between scattering intensity and scatterer distribution. Other technical revisions also strengthen the quality of the manuscript.

Regarding the novelty issue, similar to the other reviewers' opinions, the reviewer still believes that the current manuscript corresponds to a well-organized compilation of previous studies and their variations rather than presenting a completely novel approach despite the authors' response. However, based on the critical revisions mentioned above, the manuscript offers a significant contribution to the practical applications of disordered photonics, incorporating with polarization degrees of freedom and machine learning. The reviewer believes that such an extension of underlying physics to application examples meets high criteria of Nature Communications.

Therefore, the reviewer recommends the acceptance of the revised manuscript.

Reviewer #3 (Remarks to the Author):

Even the authors showed a great effort to answer the questions raised by the reviewers, most of answers were unconvincing for me. For example, my comment about the indispensable role of DM. In order to demonstrate that a DM can give different speckles, the authors used lots of words to explain the polarization state response of the DM. However, compared to SLM, the number of speckles generated by DM is quite limited, which is the main disadvantage of DM. As for the disadvantages of SLM mentioned by the authors (such as instability induced by heating under continuous power supply, calibration for different wavelength ranges, unstable performance under long-term operation, and purchase cost), they are also the disadvantages of DM! SLM is a commercial equipment, and its stability is

much higher than a metasurface. Moreover, SLM is designed for a wide wavelength range, but the DM designed by the authors need more calibrations and new DMNets training for different wavelengths. In additions, the fabrication of a metasurface is expensive, and its stability have not been tested by any long-term operation.

Another issue I agreed with Reviewer 4 is that the DM only plays a role in the encryption process but not decryption process. And this encryption process can be replaced by any computational or optical process other than a DM. This again fails to show the indispensable role of the DM. The arguments from authors are again unconvincing for me. When the authors talked about the security and duplication, they assumed the attacker can access to the encryption data. In this case, the secret information can be stolen directly, no matter which encryption methods were applied.

Agreed with Reviewer 2, I believe that the work is only a simple combination of well-established prior results in DM, deep learning and optical encryption. The quite limited novelty cannot convince me to recommend its publication in Nature Communications, even with any technical revisions from the authors.

Reviewer #4 (Remarks to the Author):

The authors have carefully addressed all my concerns and I would like to suggest its publication.

Reviewer #1 (Remarks to the Author):

(1) *I acknowledge that the authors have provided a genuine revision of their initial manuscript together with an impressive amount of detail in a rebuttal document addressing the comments by all reviewers.*

While I did write in my initial review that the publication of this work in an archival journal deserves to be considered, I am sorry to say that the whole revision work did not convince me. The rebuttal document contains many words in reply to my questions and criticism, but almost all are just repetitions of many aspects of the initial manuscript content that I had already understood. My questions have not really been addressed. My conclusion is that this work does not meet the quality standards for publication in a Nature group publication. For a brief explanation about my recommendation, I would like to highlight two points, omitting deep learning techniques.

Reply: While the reviewer claims that what the authors replied in the last response is: “almost all are just repetitions of many aspects of the initial manuscript content that I had already understood”, the authors sincerely hope that the reviewer can clarify what information being asked for by the reviewer has not been provided by the authors, to avoid any misunderstanding. Here, the authors try to briefly summarize what we had provided in the last response to the comment for the first version (in the order of the comments from the Reviewer 1 in the 1st round):

- 1 General perspective: the primary concern raised pertains to the novelty of the proposed scheme and/or the use of disordered metasurface (DM). To address this, the authors have provided additional results/discussions from three perspectives:
 - 1.1 Comparison between the one-to-one and two-to-one mapping has been given with a figure to highlight the difference regarding the decryption flow.
 - 1.2 Instead of being an information carrier as seen in many earlier studies, the DM serves as an information scrambler and a transmitter via different polarization channels, the number of which is expanded from two orthogonal polarization states (in the initial submission) to seven non-orthogonal ones (in the last response in Fig. 4 of the main text or SFig. 2 of the last Response Letter).
 - 1.3 DM as a the speckle generator provides high-level stability than a conventional ground glass, which is demonstrated through a longer-period stability test: a) in the initial submission, stability test with 72 hours was performed and included in the Supplementary, which was, however omitted as mentioned in Comment 3.5 of the first response; and b) in the last response, the stability test with more sampling periods within 80 hours is added into the main text in Fig. 5c-d.
- 2 More detailed discussion about the encryption and deciphering scheme:
 - 2.1 This study has been compared with Ref. [24]. In Ref. [24], the phase profile of the DM must be known, to which a careful alignment system has to be matched, but neither the phase profile of DM nor careful alignment system is needed in this study. Moreover, only one channel was demonstrated in Ref. [24], but multichannel encryption based on seven different polarizations has been demonstrated in this study.

- 2.2 This comment wondered why there are no comparison among different network structures. The authors did not do the comparison in the manuscript since such scheme has not been reported before and the authors chose the network structure that leads to the best performance in their practice.
- 2.3 This comment judged the necessity of the deep learning. The effective phase profile of the DM in this study is unknown because of the unknown correspondence between the DM and the input (output) and only sub-part of the speckle field needs to be recorded. There won't be a known matrix to be inverted unless the correspondence can be accessed and known in prior.
- 3 More detailed discussion about the DM as a speckle generator:
- 3.1 This comment wondered what conventional scattering medium is considered in this study. It also asked why CSM (such as a ground glass diffuser) is more unstable than the DM, which is probably related to the nature of scattering (single scattering versus multiple scattering). **More detailed discussion has been added in the reply to the 3rd comment.**
- 3.2 This comment confused the description between “deterministic” and “random” for medium. The authors had further clarified these two terms: 1) being “deterministic” in this manuscript refers that the mapping between the input and output are not changed when a medium is static or quasi-static within a temporal correlation window: the same input (e.g., the same human face with the same QR code) induces the same speckle pattern; 2) being “random” here is indeed related to the observations of the appearances of the speckle pattern arising from the scattering medium, which scrambles the input information and becomes visually uninterpretable.
- 3.3 This comment suspected whether speckles arising from a medium with “a large memory effect” is beneficial to the security. The authors have provided evidence (speckle pattern from both DM and a ground glass diffuser, and their convergence curves during the training) to show the memory effect range has no direction correlation with the security in this study and, more specifically, no significant differences occur in the network training and hence the decryption performance (before decorrelation).
- 3.4 This comment wondered if irreversible changes occur to all kinds of conventional scattering medium. The authors have attached the supplementary from Ref. [24] to cite its proof regarding the memory effect range of representative conventional scattering medium. **Nevertheless, more discussion will be included here in the reply to the 3rd comment.**
- 3.5 Stability tests comparison between the DM and the CSM, which can be referred to 1.3. **More detailed discussion has been added in the reply to the 3rd comment.**

Collectively, the authors have not merely repeated the content from the initial submission but have provided substantial results and discussions to one-by-one address the reviewer's concerns. Should these efforts be regarded insufficient, the authors sincerely request the reviewer's consideration in providing additional opportunities and specifications for further discussion.

(2) *The authors and I agree that there is nothing new about encryption methods in their work. The fact of adding complementary data to the data being encrypted (“plain text”) and merging the two in some complex operation, so that a key containing the complementary data needs to be known by the addressee is a standard technique in encryption. The fundamental contribution of this work does not lie in encryption methods.*

Reply: The authors feel deep sorry about what has been concluded from the reviewer by saying “*The authors and I agree that there is nothing new about encryption methods in their work*”, which is NOT the case at all, as we have never ever agreed that there is nothing new about encryption methods in this work.

Please allow us to reiterate the novelty in this study, which may be referred to the concern from the Comment 1. And **it is important to emphasize that no prior research in the realm of disordered photonics has explored the incorporation of complementary data (i.e., the QR code in this study) into the “plaintext”**. The absence of what is so-called “standard technique” (i.e., the double-secure scheme) in disordered photonics is due to the inherent poor stability of the environment-sensitive CSMs, which can only sustain usable performance for a limited duration of hours. This reason is **consistent** to the next point mentioned later by Reviewer #1, “a thick diffuser providing strong mixing of the ‘plain text’ data (just like added key data, in fact) is more sensitive to minute motions or environmental changes”. One of the fundamental objectives of this study is to address this issue of stability by introducing the DM as the speckle generator (also the next point raised by the reviewer and more detailed discussion will be given in the next comment). **Without addressing the stability issue, such a hardware-based encryption system cannot be achieved. Otherwise, even with a high-level security, the decryption keys (i.e., cyphertext and network parameters) held by the user need to be renewed each time when new data is added by the supplier, which is incompatible with practical decryption purpose.**

Furthermore, as stated, the contribution of this work in encryption methods does not merely depend on the double-secure scheme or any other single component like the DM. This study is aimed to **provide a speckle-based cryptosystem characterized by not only remarkable stability attributed to the robust nature of the DM, but also heightened security resulted from the double-secure scheme (preventing the brutal attack) and the expanded polarization channels (enabling channel encryption with one medium)**. Unfortunately, in the last revision, the authors have expanded the encryption channels from two orthogonal polarizations to seven non-orthogonal sets, which is entirely omitted by the Reviewer #1. This notable achievement has been **appreciated** by Reviewer #2, as quoted here, “*the manuscript offers a significant contribution to the practical applications of disordered photonics, incorporating with polarization degrees of freedom and machine learning. The reviewer believes that such an extension of underlying physics to application examples meets high criteria of Nature Communications.*”

(3) *The authors repeatedly claim that the disordered metasurfaces are more stable in time than conventional scattering materials. They also claim that they have experimentally tested that claim but they do not provide a comprehensive description of what they did, and whether it is convincing proof that the two samples (DM and CSM) were comparable in a meaningful way. If that is true, it*

deserves publication, maybe in a Nature group publication, for its own sake. That a thick diffuser providing strong mixing of the “plain text” data (just like added key data, in fact) is more sensitive to minute motions or environmental changes is not a question. To conclude, I am sorry that I do not recommend the revised manuscript for publication in Nature Communications.

Reply: While the authors have demonstrated the experimental comparison for stability analysis between a DM and a ground glass (either in the supplementary in the initial version or Fig. 5c-d in the current version) based on Comment 3.5 in the last round, the reviewer seems to have ignored these evidences. As mentioned in Comment 3.1, the reviewer thought a glass diffuser (GD) applies single scattering to photons, which is quite similar with DM, and wondered why the GD behaves in a more instable manner. Therefore, what the reviewer focused on is probably about whether or not the decryption performance exhibited by a ground GD can serve as a representative benchmark for all other CSMs, particularly given the GD’s reliance on single scattering rather than multiple scattering. In this context, let’s reiterate the rationales to compare the stability between a DM and a GD in this study.

Firstly, regarding the scattering type a GD relies on, here is a quote from Ref. [24]: “*In contrast to a conventional disordered medium (for example, a several-micrometers-thick layer of zinc oxide particles or a ground glass diffuser) that has three-dimensional optical inhomogeneity (that is, the thickness of the scattering medium is much larger than the optical wavelength), the disordered metasurface is composed of a two-dimensional array of subwavelength scatterers of uniform height.*” The cited sentence indicates that the GD they used for demonstration relies on multiple scattering. It is worth noting that the GD model in Ref. [24] and our study are DG-10-120 (Thorlabs) and DG-10-220 (Thorlabs), respectively, whose fabrication methods are the same from the same manufacturer, with different grit sizes though. Therefore, it is rational to claim that the GD used in this study relies on multiple scattering. Additionally, the type of scattering varies medium by medium, depending on the material, structure, geometry, and fabrication defects, which is however beyond the scope of this study. And what the encryption system needs is to scramble the spatial distribution of the input plaintext so that no clear features can be observed in the output field; a scattering medium of high stability is therefore desired as a speckle generator, regardless of the scattering type.

Secondly, in the pursuit of stable performance, a medium with large ME range is preferred, and a DM is the ideal choice since it possesses much larger ME range when compared to the GD that has largest ME range among representative CSMs, as demonstrated in Ref. [24]. More specifically, as shown in SFig. 1a (Supplementary Figure 2b in Ref. [24]), a larger ME range of the medium indeed indicates slower decorrelation regarding the increasing tilted angle of the medium, i.e., background

PCC drops to 0.5 (or 0.8) at the angle of 6° (or 2.5°) and 30° (or 11°) for GD and DM, respectively. DM is subjected to weaker decorrelation under the same perturbations and is significantly less sensitive to the external influences as compared to GD. Further, let's address two terms “instable” and “irreversible change” for the GD as indicated by the reviewer in the Comments 3.1 and 3.4 in the last round of review. In this study, the medium status is randomly perturbed due to many daily factors for a long period of time. Thus, the medium status is changed according to a general tendency to reduce the background PCC (decorrelation) with fluctuations. Larger ME range therefore leads to two benefits:

- 1) **Insensitive to perturbations:** the decorrelation of DM becomes less sensitive to perturbations, and the DM even seems to be static or quasi-static, compared to the GD, which can be found in Fig. 5 in our manuscript: the performance with DM remains excellent and stable for more than 120 hours regardless of the external perturbations, while that with CSM (i.e., a GD) quickly declines in around 2 hours and the recognizable features in the decrypted faces fade with the data collected 2 hours later than the initial period.
- 2) **Easier to “reverse”:** when decorrelation occurs, the increasing perturbations will first break a smaller ME range and then a larger one. If we define effective decryption PCC as 0.9, the effective decryption range regarding background PCC should be above 0.8 (decryption PCC are generally above 0.9 when background PCC is above 0.8 as shown in Figs. 5a&c). As shown in SFig. 1b below, only a small probability occurs when the medium status is around the effective decryption range of GD (Region A) while much larger one occurs when the medium status is in between the effective decryption ranges of DM and GD (Region C). Therefore, even with fluctuations that change the medium status back and forth, the statuses of S_2 - S_4 can be observed much more easily than those of S_0 - S_1 . As a result, the status of DM seems to be reversible (i.e., recovered background PCC occurring in Periods 2 ~ 3, Periods 7~8, and Periods 12~13 in Fig. 5a), but that of GD seems to be “irreversible” (i.e., general decorrelation with fluctuations).

SFig. 1 (a) Speckle decorrelation for different scattering media including DM (the blue points) and three other CSMs (ground glass, white paint, and opal glass). (b) Probability of the medium status changing in effective decryption

ranges of DM and GD. Figure (a) is reproduced with decorrelation curves from the supplementary of Ref. [24]. DM: disordered metasurface; GD: ground glass diffuser.

Collectively, **the speckle decorrelation and/or its corresponding medium status can be changed by the random perturbations back and forth, which is reversible, but wider ME range makes the reversibility easier. Equivalently, with larger memory effect range, larger relative motion can be tolerated by the decryption system and hence stronger noise-resisting ability and more stable performance with the DM.** This characteristic is believed to guarantee the feasibility of achieving a reliable encryption system with DM, since no additional calibrations (i.e., DMNet training in this study) are needed with new encrypted data. Such stability therefore provides promising feasibility towards the practical applications as an encryption system. The majority of published studies related with speckle-based explorations tend to avoid or ignore addressing this point, except Resisi's work [Laser & Photonics Review 15(10), 2000553 (2021)] that let the neural network to learn from data covering as many as statuses of the CSM (equivalent to calibration with new data indeed). Even so, the performance there is poor even for simple information like handwritten digits/letters, which has limited the feasibility of wide applications for that study.

Reviewer #2 (Remarks to the Author):

In this resubmission, the authors provided a thorough revision in response to the reviewers' comments. The revised manuscript highlights the following critical points. First, the authors showed that the suggested platforms operate not merely with two-channel orthogonal polarization but with their superposition, resulting in an enhanced information capacity. Second, the authors established the validity of their method to a specific yet broad class of problems, skillfully evading an issue related to one-to-many relationship between scattering intensity and scatterer distribution. Other technical revisions also strengthen the quality of the manuscript.

Regarding the novelty issue, similar to the other reviewers' opinions, the reviewer still believes that the current manuscript corresponds to a well-organized compilation of previous studies and their variations rather than presenting a completely novel approach despite the authors' response. However, based on the critical revisions mentioned above, the manuscript offers a significant contribution to the practical applications of disordered photonics, incorporating with polarization degrees of freedom and machine learning. The reviewer believes that such an extension of underlying physics to application examples meets high criteria of Nature Communications. Therefore, the reviewer recommends the acceptance of the revised manuscript.

Reply: We deeply appreciate the recognition of the reviewer regarding the revision and explanation of the work.

Reviewer #3 (Remarks to the Author):

1. (a) *Even the authors showed a great effort to answer the questions raised by the reviewers, most of answers were unconvincing for me. For example, my comment about the indispensable role of DM. In order to demonstrate that a DM can give different speckles, the authors used lots of words to explain the polarization state response of the DM. However, compared to SLM, the number of speckles generated by DM is quite limited, which is the main disadvantage of DM. (b) As for the disadvantages of SLM mentioned by the authors (such as instability induced by heating under continuous power supply, calibration for different wavelength ranges, unstable performance under long-term operation, and purchase cost), they are also the disadvantages of DM! SLM is a commercial equipment, and its stability is much higher than a metasurface. Moreover, SLM is designed for a wide wavelength range, but the DM designed by the authors need more calibrations and new DMNets training for different wavelengths. In additions, the fabrication of a metasurface is expensive, and its stability have not been tested by any long-term operation.*

Reply: (a) In the last round, the reviewer proposed that “Instead of DM, SLM can also be used to generate speckles in the optical encryption scheme, which can further enhance the stability. Moreover, SLM can be a dynamic one to generate different patterns for the encryption”. In the last response letter, we think we have given sufficient discussion in terms of these two concerns. Nevertheless, in this round, we would like to take this chance to give more detailed discussions.

Regarding stability, we do agree that the SLM can also enhance the stability since its pixel size is relatively large (~6 μm) without considering the influence of continuous accumulation of heat, which can tolerate some environmental vibrations applied to the system. For DM, the periodic constant of meta-pillar in this experiment is 350 nm. The effective size of the unit can be enlarged by merging several adjacent meta-pillars with the same phase response. By doing so, the DM-based system can be also further enhanced in principle. Yet, based on our experimental results as shown in Fig. 5 in the revised manuscript, the DM-based system has shown featured stability over the CSM-based system. The stability is thus sufficient for the purpose of optical encryption due to the single layer structure of the DM, ensuring excellent temporal generalization of the encryption system and avoiding additional training for the networks when new data is added. This finding is consistent with what is reported in the Result section of Ref. 24 [Nat. Photon. 12, 84-90 (2018)]: “Moreover, the disordered metasurface is extraordinarily stable due to its fixed, two-dimensional fabricated structure. We were able to retain the ability to generate a high-quality optical focus from the same metasurface without observable efficiency loss over a period of 75 days by making only minor corrections to the system alignment to compensate for mechanical drift (see Supplementary Fig. 3)”. These (experimental results from literature and our experiments) can confirm the stability of the DM to ensure long-term operation.

A SLM, as an active component, can inherently serve as a dynamic speckle generator, but it requires continuous power supply and has a bulky size. In this work, the DM can, without power supply, serve as an actively tunable speckle generator in a compact and passive manner by simply adjusting the incident polarization. That is the reason why we “used lots of words to explain the polarization state response of the DM”. Regarding the size, a SLM indeed sees limitations due to its bulky size. If one wants to duplicate the encryption system elsewhere, it is believed that taking a key of small size (i.e., one DM or even thousands of DMs) is much safer and more convenient than taking one SLM; the bulky size of a SLM and its protection (inevitable as a buffer for any mechanical crash) are emphasizing its importance and exposing itself to the potential hackers. Considering the dynamic speckle generation or the number of possible speckles, in this experiment, we demonstrated seven independent channels with seven non-orthogonal incident polarizations by simply rotating the quarter-wave plate with axis angle varying from 45° to 135° with a step size of 15° , as a proof of concept. By jointly adjusting the half-wave plate and the quarter-wave plate, more polarization channels can be created, where the induced speckle pattern varies channel by channel with the same input information (Fig. 2c). Also, our cross-validation for the multichannel decryption in Fig.4c-d shows only when the polarization of input speckle pattern matches that of the DMNet can pass the decryption, indicating that the speckle patterns from different polarizations can be well-differentiable by the decryption network. In principle, arbitrary polarization state could be used as an encryption channel. The comment “*the number of speckles generated by DM is quite limited, which is the main disadvantage of DM compared to SLM*” may not represent the real scenario in the context.

(b) Here we want to clarify that instability induced by heating under continuous power supply and unstable performance under long-term operation are associated with the unstable phase response of the elements in SLM, but not the stability induced by the environmental perturbation. This issue does not exist for DM, as it functions passively, and no power is needed to drive it as the speckle generator.

Referring to the cost, the price of a state-of-the-art commercial SLM package (including the screen and the control box) is at the scale of **10,000** US dollars and, of course, it fluctuates with manufacturers and models. In comparison, the fabrication cost of a metasurface using standard nano-fabrication technique is at the scale of **100** US dollars per sample in our case. This is much cheaper than a commercial SLM. Recently, the commercialization of metasurfaces prompted by some companies such as Metalenz, Inc. (<https://metalenz.com/>) could further reduce the cost.

The phase response of a SLM for different wavelengths are different and hence the speckles for different wavelengths are different and new network training is also required for additional wavelengths. In terms of the range of the electromagnetic spectrum, the concept using DM is

applicable over a wide range of the electromagnetic spectrum with specific design, from ultraviolet to terahertz light with a proper choice of low-loss materials for the meta-atoms [Sci. Adv. 8, e5644 (2022); Advanced Photonics 3, 036003 (2021); Nat. Commun. 12, 5560 (2021)]. On the contrast, the available wavelength range of a commercial SLM is usually from 350 nm to 1700 nm (<https://holoeye.com/products/spatial-light-modulators/>) and, to cover wider spectrum, more SLMs are technically needed, which will again significantly increase the cost.

Some published metasurface related work can also support our arguments especially in terms of the cost and component size. For example, in the reference [Adv. Mater. 34, 2109714(2022)], the statement “...However, these beams are mainly generated using spatial light modulators, which suffer from large volume, high cost, and limited resolution. Benefiting from the ultrathin nature and unprecedented capability in light manipulation, optical metasurfaces provide a compact platform to perform this task...” is provided in the **Abstract**; in the reference [ACS Photonics 10(7), 2045–2063(2023)], the comparison between SLM and metasurface “Conventionally, wavefront shaping of light relies on the accumulation of optical paths in optical systems, which brings bulky volume and integration challenges to optical component design. Spatial light modulators (SLM), which utilize the electro-optic properties of liquid crystals to modulate light, have shown the advantages of dynamic and reconfigurable functionalities. However, owing to the large pixel size, the field of view (FoV) and modulation efficiency are limited and unwanted diffraction orders are unavoidable. Meanwhile, the integration of SLM is expensive and complicated in real-world applications. In comparison, metasurfaces can generate abrupt phase changes within ultrathin thicknesses, thereby providing compact and efficient solutions for wavefront engineering” is concluded in “**Wavefront Shaping Based on Metasurfaces**” section.

2. Another issue I agreed with Reviewer 4 is that the DM only plays a role in the encryption process but not decryption process. And this encryption process can be replaced by any computational or optical process other than a DM. This again fails to show the indispensable role of the DM. The arguments from authors are again unconvincing for me. When the authors talked about the security and duplication, they assumed the attacker can access to the encryption data. In this case, the secret information can be stolen directly, no matter which encryption methods were applied.

Reply: The reviewer used Comment #2 from Reviewer #4 in the first round as a new argument, which is indeed involved a more general comparison between the hardware- and software-based encryption approaches but not specifically for the DM realizations. Please note that in this round of review, Reviewer #4 has accepted our explanation and the corresponding revision. The author would like to further emphasize the benefits of our proposed DM-based scheme as a complementary to the response to the Comment #2 from Reviewer #4 in the last round.

To crack a cryptosystem, mainly three approaches are of interest, including 1) to duplicate the forward process (i.e., encryption system) and then solve the inverse process (i.e., decryption); 2) to steal both plaintext and cyphertext data and to figure out the intrinsic relationship among plaintext

and cyphertext for decryption; 3) brutal attack. The brutal attack will be out of discussion due to its involvement with decryption process only and it has been demonstrated in Fig.4a, so that only aspects 1) and 2) will be discussed here to justify the concern from the reviewer:

- 1) The hardware encryption system can be performed offline in an isolated environment, so that **the system should be inaccessible to the external users or hackers**. Even if the hackers get the phase profile of DM, they cannot reproduce and duplicate the encryption system without knowing the precise alignment of the system and hence fail to solve the inverse process. Also, the processing speed greatly benefits from the speed of the light, regardless of the high complexity and high dimensionality (long key length) of the DM. For software encryption, the processing speed and the key length (longer length give higher security) are however needed to be compromised.
- 2) Regarding the access to data, this may be referred to the claim from the reviewer: “When the authors talked about the security and duplication, they assumed the attacker can access the encryption data. In this case, the secret information can be stolen directly, no matter which encryption methods were applied”. It seems to be a misunderstanding of our explanation from our point of view. In the last response letter, we have explained that: *“...The physical encryption process will be digitally inaccessible and hard to be duplicated (more detailed discussion is given in the next paragraph), even if an attacker knows the phase profile of the metasurface. By doing so, even if the attacker has access to the encryption data (as the data needs hard disk for storage or network for transmission), he will fail to decrypt or understand the data...”*. The encryption data indicate the ciphertext (speckle) which is the output from the physical encryption process from the context in this response. **The secret information termed by the reviewer is plaintext, which can be deleted from the hard disk right after the DMNets have been trained. Therefore, the intrinsic relationship between the plaintext and ciphertext can hardly be guessed since only the ciphertext can be probably stolen from the hackers and no access is presented to the plaintext.** In our last response to Comment #2 from Reviewer #4, the secret information (plaintext) cannot be decrypted or understood without accessing the physical encryption process.

Compared with other optical processes, especially in disordered photonics, **only a single DM can serve as a tunable speckle generator in a passive manner with a function of multiple polarization encryption channels with one medium. As a result, the information transmission channel and security are greatly improved using a spin-multiplexing DM.** Furtherly, to support such high-level security, stable encryption performance is a “must”, hence the speckle decorrelation MUST be addressed. DM has so far been reported as the most promising solution as a medium to

address the speckle decorrelation. The DM with large memory effect range can effectively tolerate external perturbations and maintain the same speckle pattern with the same input for a long time, i.e., 75 days as reported by Ref. [24] and >140 hours as demonstrated by our study, while conventional scattering media (CSM) can merely stand for ~ 1 hour. The authors therefore believe that such significant contributions from the DM are indispensable for an encryption system towards practical applications. **Otherwise, even with high-level security, the decryption keys (i.e., cyphertext and network parameters) held by the user inevitably need to be renewed or updated each time when new data is added by the supplier, which is incompatible with practical applications.**

3. *Agreed with Reviewer 2, I believe that the work is only a simple combination of well-established prior results in DM, deep learning and optical encryption. The quite limited novelty cannot convince me to recommend its publication in Nature Communications, even with any technical revisions from the authors.*

Reply: The reviewer uses Comment #1 from Reviewer #2 in the first round as another new argument. In this round of review, however, Reviewer #2 has also accepted our revision. Nevertheless, we would like to give more explanations to clarify the confusion.

In this study, the employment of DM as a stable and tunable speckle generator in a passive manner coupled with a double-secure procedure contributes to a highly secure speckle-based cryptosystem towards great promise in practical applications. To attain the purpose, three features, including ultra stability of DM, double-secure procedure, and a tunable speckle generator with DM, are utilized.

Although the stable feature of DM has been previously validated in Ref. 24 [Nat. Photon. 12, 84-90 (2018)], in this study ultra stability is the prerequisite condition to allow for the immunity of additional calibrations for new data. Otherwise, even with high-level security, the decryption keys (i.e., cyphertext and network parameters) held by the user inevitably need to be renewed each time when new data is added by the supplier, which is not desired or allowed in applications. The stability is the very core that this study tries to address by introducing DM.

The security is always the most important parameter in a usable cryptosystem. A double-secure procedure and tunable speckle generator in a passive manner using DM with polarization degrees of freedom are proposed to boost the security in two different dimensions in this study.

In learning-based disordered photonics, the idea of a double-secure procedure (adding security key, i.e., the QR code in this study) applied on the “plaintext” to increase the complexity of the correspondence between the input and the output of the system has not been reported before. This

is because, without addressing the stability issue, such hardware-based encryption system cannot be achieved in experiment. Although the double-secure procedure seems “simple”, it truly changes the mapping relation between the input and the output of the physical encryption system, with which the system gains effective protection from brutal attacks (Fig. 4a). Moreover, the security key can be altered in real time, which can greatly enhance the security of the cryptosystem.

Compared with other works relying on spin-multiplexing metasurfaces, the polarization sensitive DM in this work exhibits very different characteristics. In previous works, the information is typically encoded within the metasurface structures, serving as physical carriers that support limited channels (normally two channels with orthogonal polarization states) for effective information transmission. Under other incident polarizations other than the predesigned polarizations, the transmitted information is invalid due to crosstalk. In our work, the information is encrypted onto the output speckles instead of relying solely on the metasurfaces themselves, and in the meanwhile these output speckles under different incident polarizations are different from each other. In the last revision, we have demonstrated seven independent polarization channels between two orthogonal polarization channels, in comparison with only two orthogonal polarization channels in the original version, to fully support our argument of infinite polarization channels in the original manuscript. In principle, the number of channels associated with the incident polarization is infinite by simply adjusting the polarization of the incident light. As a result, the information transmission channel is dramatically increased with such a polarization sensitive DM. In other words, a single DM serve as an actively tunable speckle generator which has never been demonstrated by other DM or metasurface related works.

To conclude, the DM with spin-multiplexing feature in this study provides an ultra-stable platform that supports multiple polarization encryption channels, leading to a highly secure speckle-based cryptosystem towards great promise in practical applications. The work is fundamentally different from other DM-based works regarding the principle and application scenario. For each encryption channel, the double-secure procedure will furtherly increase the security in a different dimension. Therefore, our work is not a simple combination of well-established prior results in DM, deep learning, and optical encryption.

Reviewer #4 (Remarks to the Author):

The authors have carefully addressed all my concerns and I would like to suggest its publication.

Reply: We appreciate the recognition of the reviewer regarding the revision and explanation.

REVIEWERS' COMMENTS

Reviewer #1 (Remarks to the Author):

I am sorry that I am not able to contribute to this new reviewing iteration. The authors have minimally revised their manuscript and have embarked in a lengthy and detailed discussion about why they disagree with my previous comments, while some of the other reviewers think that this article is now acceptable for publication in Nature Communications. I am not ready to contribute to this kind of competition and I just would like to express my thanks to the authors for the time it took them to write their rebuttal letter. Let me avoid standing on the way and objecting to publication of this work.

Reviewer #3 (Remarks to the Author):

I am sorry to say that the authors' reply cannot convince me. Although the authors repeat their claims again and again, I do not believe the arguments are convincing. In my opinion, the proposed method conveys nothing new for the optical encryption. The disordered metasurface fails to show its indispensable role in the encryption scheme. Therefore, I do not recommend the revised manuscript for publication in Nature Communications.

Response Letter

REVIEWERS' COMMENTS

Reviewer #1 (Remarks to the Author):

I am sorry that I am not able to contribute to this new reviewing iteration. The authors have minimally revised their manuscript and have embarked in a lengthy and detailed discussion about why they disagree with my previous comments, while some of the other reviewers think that this article is now acceptable for publication in Nature Communications. I am not ready to contribute to this kind of competition and I just would like to express my thanks to the authors for the time it took them to write their rebuttal letter. Let me avoid standing on the way and objecting to publication of this work.

Reply: Thanks a lot for the reviewer and his/her insightful comments given in all cycles, which have greatly helped to improve this manuscript and our understanding on the topic.

Reviewer #3 (Remarks to the Author):

I am sorry to say that the authors' reply cannot convince me. Although the authors repeat their claims again and again, I do not believe the arguments are convincing. In my opinion, the proposed method conveys nothing new for the optical encryption. The disordered metasurface fails to show its indispensable role in the encryption scheme. Therefore, I do not recommend the revised manuscript for publication in Nature Communications.

Reply:

Although the reviewer still insists his/her criticism regardless of the authors' response, he/she is highly appreciated for the comments that have triggered us to think and express more critically. Nevertheless, the concerns of the reviewer have been addressed in detail regarding why the metasurface is indispensable in this study from different aspects. The authors are, likewise, standing that our demonstration after several rounds of revision is convincing. And therefore, the authors have to disagree with this part of the assessment. That said, the authors still respect the expertise of the reviewer and are sincerely grateful for the efforts from the reviewer to help us to improve this manuscript.